# Interference Challenges and Management in B5G Network Design: A Comprehensive Review

**Osamah Thamer Hassan Alzubaidi [1], MHD Nour Hindia [1], Kaharudin Dimyati [1,*], Kamarul Ariffin Noordin [1], Amelia Natasya Abdul Wahab [2], Faizan Qamar [2,*] and Rosilah Hassan [2]**

[1] Department of Electrical Engineering, Faculty of Engineering, University of Malaya, Kuala Lumpur 50603, Malaysia

[2] Centre for Cyber Security, Faculty of Information Science and Technology (FTSM), Universiti Kebangsaan Malaysia (UKM), Bangi 43600, Selangor, Malaysia

[*] Correspondence: kaharudin@um.edu.my (K.D.); faizanqamar@ukm.edu.my (F.Q.)

**Abstract:** Beyond Fifth Generation (B5G) networks are expected to be the most efficient cellular wireless networks with greater capacity, lower latency, and higher speed than the current networks. Key enabling technologies, such as millimeter-wave (mm-wave), beamforming, Massive Multiple-Input Multiple-Output (M-MIMO), Device-to-Device (D2D), Relay Node (RN), and Heterogeneous Networks (HetNets) are essential to enable the new network to keep growing. In the forthcoming wireless networks with massive random deployment, frequency re-use strategies and multiple low power nodes, severe interference issues will impact the system. Consequently, interference management represents the main challenge for future wireless networks, commonly referred to as B5G. This paper provides an overview of the interference issues relating to the B5G networks from the perspective of HetNets, D2D, Ultra-Dense Networks (UDNs), and Unmanned Aerial Vehicles (UAVs). Furthermore, the existing interference mitigation techniques are discussed by reviewing the latest relevant studies with a focus on their methods, advantages, limitations, and future directions. Moreover, the open issues and future directions to reduce the effects of interference are also presented. The findings of this work can act as a guide to better understand the current and developing methodologies to mitigate the interference issues in B5G networks.

**Keywords:** B5G; interference; HetNet; D2D; UDN; UAV

## 1. Introduction

The rapid growth and sustained advancement in future wireless technologies of different new applications in Beyond Fifth Generation (B5G) communication networks have led to a massive growth increase in demand for user data. The volume of mobile data traffic was 7.462 EB/month in 2010 and it is expected that this traffic will be 5016 EB/month in 2030 [1]. A massive number of surveys were conducted and are continuously being performed in different areas of wireless communication such as interference management, mobility management, spectrum management, and energy management [2–5]. Researchers from various portions of networking and communication institutions, from academics to marketers and providers to operators, have collaborated and introduced an effective way to make this subject possible [6]. This development resulted in an increased growth rate in the user data rate of around 10 Gb/s with a minimum latency of around <1 ms, mobility of >1000 km/h, reliability of 99.999%, and better battery lifetime [7,8]. Furthermore, this trend is expected to increase dramatically during the next few years. The collection of advancements in previous technologies and innovative wireless transmission technology demonstrates a tendency to meet cellular users' requirements and the objectives of the next generation such as B5G, also known as the sixth generation (6G) networks [9,10]. Usually, the substantial rise in traffic is a direct result of an increase in request for certain services, such as super-intelligent society (SIS) [11], extended reality (ER) [12], connected robotics

integrated systems [13], wireless interactions between computer and brain (WICB) [14,15], haptic communication (HC) [16], smart healthcare and biomedical communication [11], automation and manufacturing, information transfer through the five senses, internet of everything (IoE) [17–19], etc. Researchers also anticipate the better Quality of Service (QoS) needs for massive data and real-time applications while maintaining secure communications [20].

The combination between millimeter-wave (mm-wave) and THz bands was inserted in existing B5G networks due to the restrictions of the previous mobile generation and the requirement for wide bandwidth [21]. This provides a large amount of unused frequency bands that can be used to enhance the system spectral efficiency (SE) approximately by seven times compared with traditional homogeneous networks [22]. The combined frequency range depends on a direct transmission in order to mitigate the path loss for the cellular users (CUs) in both Line-of-Sight (LOS) and Non-Line-of-Sight (NLOS) transmission [23]. However, because of the short wavelength as well as several intra-cell interferences by neighboring devices, the frequency bands of B5G suffer from attenuation, fading, reflection, refraction, scattering, shadowing, and absorption by brick walls and concrete buildings, preventing them from penetrating long distances [24,25]. Meanwhile, in ultra-dense network urban areas, users face penetration, scattering, and interference problems. Therefore, it is very important to model the channel of each spectrum before the actual implementation [26]. Because of this, the weak signals received by edge cell user equipments (UEs) are compared with the minimum required communication signal. Small base station (SBS) deployment within the cell to improve the strength of the signal is not regarded as an applicable solution because it boosts inter-cell interference (ICI). This requires a highly complex coordination scheduling (CS) algorithm which can be costly to create [27–29].

In today's world, communication technologies are key to political, economic, and socio-cultural evolution [30,31]. The interference in wireless communication technologies has become the main issue for service providers by decreasing the QoS and limiting the advantages derived from this crucial technology, leading to a decrease in revenue [32,33]. Therefore, interference is the most significant factor that influences capacity and the QoS provided to end-users [34,35]. To reduce collisions and time wastage on retransmissions, the system must be able to detect potential interferences in the spectrum and smartly utilize the spectrum [36]. The first stage toward attaining this objective is to identify and classify the existing types of environmental interferences.

The most common kinds of interference incorporated with cellular networks are self-interference, inter-user interference, adjacent channel interference, multi-access interference (co-channel interference and multi-user interference), inter-carrier interference, inter-channel interference, intra-channel interference, inter-symbol interference, inter-numerology interference, cross-link interference, inter- and intra-beam interference, multiple-access interference (intra-cell interference and inter-cell interference). Nevertheless, these types of interferences are not the only ones that affect wireless communication networks. Each network is influenced by interference incurred by its deployment and transmission scenario. The main aim of this paper is to explore and highlight the issues of interference that were noticed in various structures and approaches of the B5G network, primarily concentrating on HetNets, Device-to-Device (D2D), Ultra-Dense Networks (UDNs), and Unmanned Aerial Vehicles (UAVs), to acquire novel insights into constructing efficient B5G networks as fast as possible.

The remainder of the paper is organized as follows. Section 2 highlights the research motivation and contribution of this work. Section 3 discusses the multiple interference effects in various system models such as HetNet, D2D, UDNs, and UAVs. This includes the scenarios, benefits, interference types, and recent related work for each model. In Section 4, the open issues and future directions are provided. Finally, Section 5 concludes the paper. The structure of this paper is shown in Figure 1.

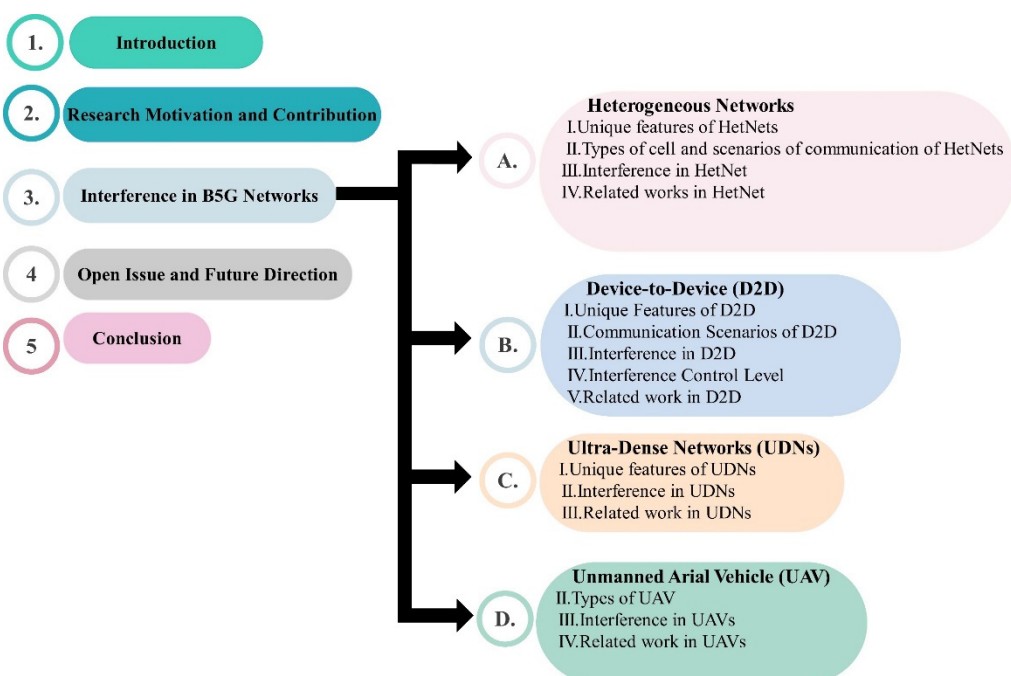

**Figure 1.** The organization of the Paper.

## 2. Research Motivation and Contribution

In the future wireless cellular networks, such as B5G, several interference problems have existed due to mobility of users, deployment of SBS with RNs that have low power and random deployment, unpredicted channel variation, sharing the frequency spectrum, utilization of frequency reuse concept, and the effect of intra-cell and inter-cell interferences [37,38]. Therefore, this interference represents the main challenge for the new frequency spectrum of B5G cellular networks.

Interference management issues have become the concern of many researchers in recent years, therefore, different technologies have been discovered to reduce various kinds of interference either by eliminating or coordinating them. However, ICI has the highest effects on multi-tier cellular networks. ICI is classified into two types, static coordination such as inter-cell interference coordination (ICIC) or dynamic coordination such as enhanced ICIC (eICIC). To eliminate the ICI, ICIC permits the UEs at the cell edge in adjacent cells to utilize various ranges of frequency such as resource blocks (RBs) and sub-carriers [39]. Furthermore, eICIC permits UEs to utilize various subframes (time ranges) for the same objective. In other words, a macro-cell (MC) and small-cell (SC) sharing a co-channel can utilize radio resources by various subframes [40]. These interference issues become more severe in B5G, resulting in a significant drop in the total performance of a network. The following points summarize the major contributions of this work:

1. In the research community, proper and robust techniques for canceling interference and then lowering the noise level are still required. Therefore, we contribute to the current literature on the interference mitigation techniques in B5G by providing new insights into the management of interference issues in future generation networks.
2. To the best of our knowledge, the interference in HetNets in the literature is focused on co-tier interference and cross-tier interference. Moreover, in the D2D, the authors usually focus on power allocation and spectrum allocation strategies. However, limited attention was paid to the hybrid interference in these environments. The current study adds the issues related to hybrid interference to the literature. This provides a new vision for researchers to mitigate the interference issue in B5G.
3. The interference in the UDNs in this study is extended to include the spatial domain. This helps the researchers in the investigation of techniques toward higher overall user performance.

4. The different types of interference from UAVs were discussed in detail including drone interference, which is rarely mentioned in the literature.
5. In this work, future research challenges and suggested methods to reduce interferences are also covered.

## 3. Network Architecture of B5G for Reducing the Interference

The new network architecture is a combination of several technologies, including HetNet, D2D, UDNs, UAVs, beamforming, Massive Multiple-Input Multiple-Output (M-MIMO), mm-wave, etc. Architectural improvements are necessary to ensure that the new radio is compatible with a conventional network. However, diverse technology models' design and contemporary practice result in massive interference in each other's signals. These susceptible interferences affect the performance of the entire network [41,42].

From a technological perspective, B5G will merge terrestrial wireless communication, satellite communication, and direct communication over short distances. Concurrently, B5G will incorporate communication, perception, computing, navigation, and other emerging technologies. By leveraging the management of intelligent mobility and the methods of control, B5G will create a new architecture of the 3D core network that can combine those systems and assist universal ubiquitous coverage of very high-speed communications, comprising communications on the earth, in space, in air, and over the sea [7,43]. Figure 2 shows the architectural advancements used in B5G networks [23].

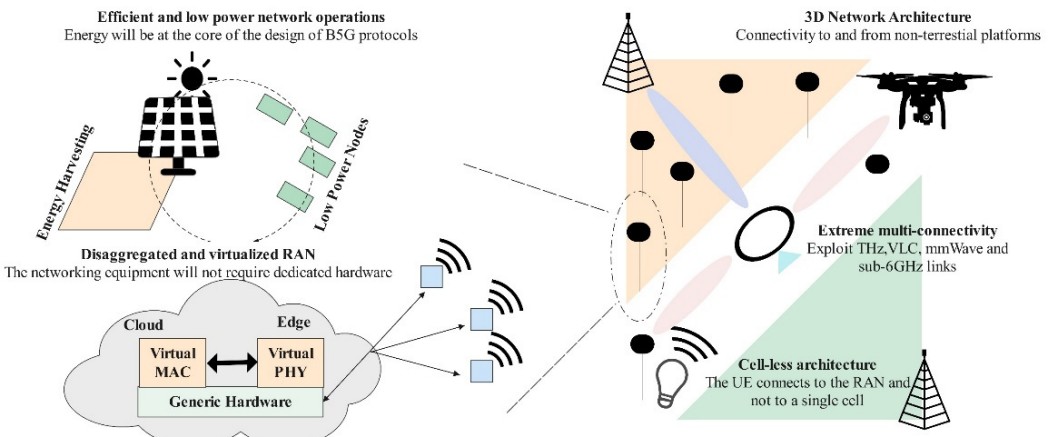

**Figure 2.** The architectural advancements in B5G networks.

Although this 3D core network design has the potential to overcome conventional coverage restrictions and eventually establish an unparalleled universal coverage, some challenges should be solved to improve the performance of the B5G networks [21]. Interference is the most significant factor that influences the capacity and QoS provided to end-users. Therefore, in the B5G network, it is essential to explore how interference can be canceled using traditional interference cancellation techniques, such as successive interference cancellation (SIC) and parallel interference cancellation (PIC), or key enabling technologies such as M-MIMO, intelligent beamforming (IB), resource allocation (RA), etc. [1,44].

## 4. Interference in B5G Networks

This section reviews the multiple interference impacts in HetNet, D2D, UDNs, and UAVs. It illustrates the outcomes of the latest studies to explain different types of interferences associated with each method. Figure 3 illustrates the interference in B5G networks.

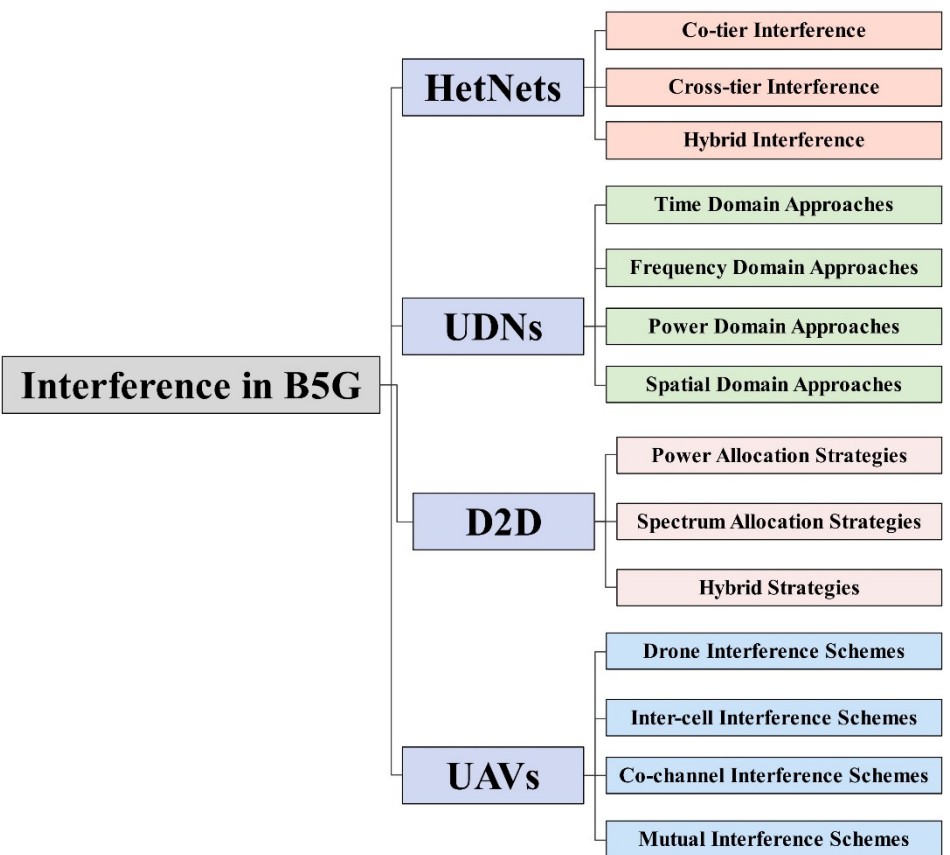

**Figure 3.** Interference in B5G networks.

*4.1. Heterogeneous Networks*

The future of B5G wireless networks is forecasted to manage applications demanding massive data rates. One of the proposed solutions given by the third generation partnership project (3GPP), Release 12, is to meet data rates demand to allow network densification through the deployment of SCs [45]. Such densification provides increased spectrum efficiency, increased network capacity and overall performance, cost-effective coverage expansion, and can even lower the mobile's power consumption because of communication with surrounding pico-cells [46]. This solution increases the coverage of the network dramatically. However, it necessitates invention in hardware miniaturizing and cost reduction in the construction of a small-cell base station (BS). These SC-BSs can be installed as low-power femto-cells for enterprise or residential installations or as higher-power pico-cells to improve a macro-cell's outdoor coverage. The synchronous operation of macro-, micro-, pico-, and femto-cells is referred to as HetNets [47]. Particularly, HetNet enables different types of SCs to cohabit with macro-cells by participating in the same resources of the spectrum, which can significantly enhance SE and decrease uncovered regions [48].

4.1.1. Unique Features of HetNets

1.  Increase the capacity of the system: By allowing many mobile terminals with varying access technologies to cohabit in the same physical location, the total system capacity can be considerably increased.
2.  Massive density: To provide ultra-connectivity, multiple users with varying levels of power are distributed by deploying many SCs. The structure of the network gets significantly denser.
3.  Reduction of uncovered regions: With the deployment of diverse SCs (e.g., micro-cells, femto-cells), it is possible to decrease uncovered regions and extend the range of communication by improving access points in the environment of the poor channel.

4.  Decrease path losses and delay: In a wide-region communication environment without SCs, the channel path loss between mobile terminals and macro base station (vs) is severely deteriorated due to the vast distance between various devices. While slight path losses can be faced to the backhaul signals from mobile terminals to MBSs when SBSs are located between MBSs and mobile terminals [49].

5.  Increase SE: Given the scarcity of available spectral frequencies in traditional homogeneous networks, it is preferable to discover an efficient way to increase the SE of the system [50]. The radio frequency (RF) unit must be redesigned when the radius of transmission is small in the high band of frequency. However, HetNet can increase the SE and enable smooth connection at any time and everywhere by cohabiting with diverse cells, as shown in Figure 4. The figure depicts how various networks with various functions are divided into different tiers that span from space to ground communications. Particularly, the conventional HetNet is a depiction of terrestrial communications, such as macro–micro HetNets. By participating in the authorized spectrum with MC users, various emerging networks (e.g., D2D, vehicle-to-vehicle (V2V)) and conventional macro networks are combined into a multi-tier HetNet. However, the future directions indicate that HetNets via terrestrial communications will be developed toward space communications, such as communications at low altitudes and communications in deep space. For example, the spectrum of ground stations can be shared by D2D users when the UAV acts as an air BS serving different ground stations, forming a heterogeneous coalition network with low altitudes. Additionally, a spatial HetNet can be formed by balloons, satellites in deep space, and satellites in near orbit.

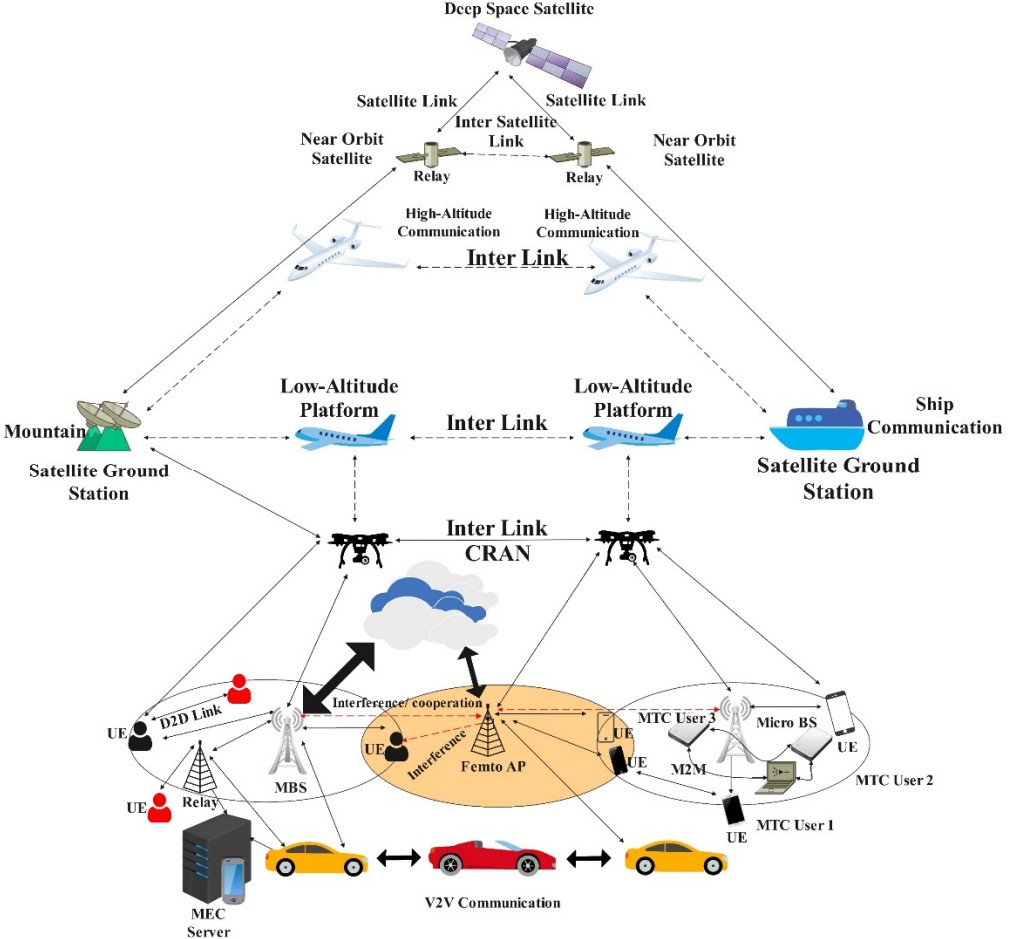

**Figure 4.** The structure of future HetNets.

4.1.2. Types of Cell and Scenarios of the Communication of HetNets

Figure 5 shows a perfect HetNet with various SCs. The network types can be classified into four categories based on their various coverage areas and operation scenarios [51–53] as follows.

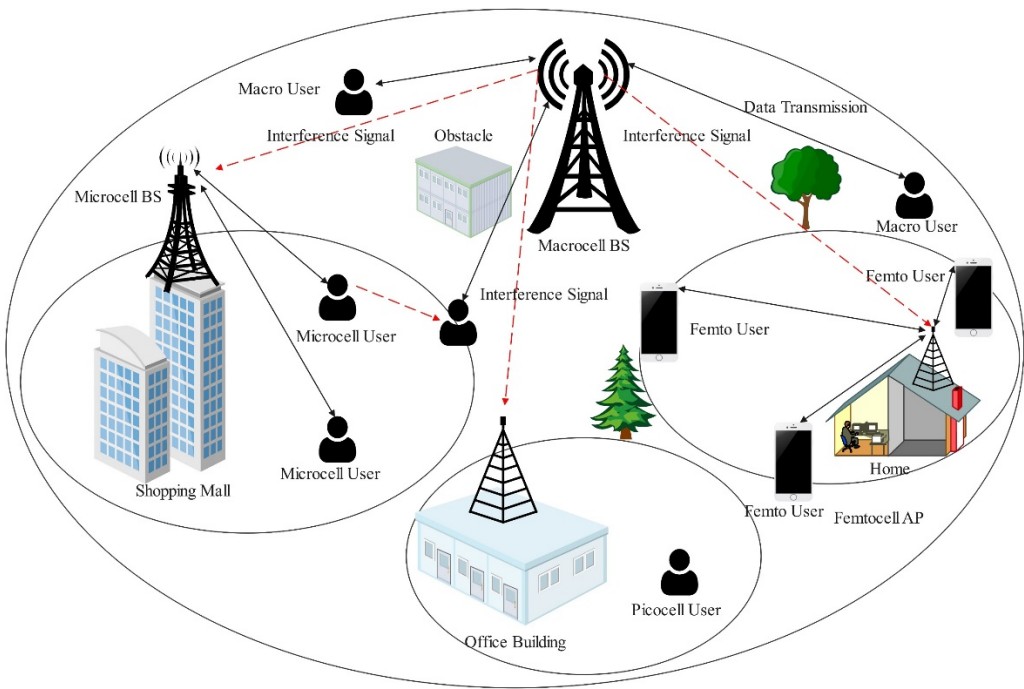

**Figure 5.** The two-tier HetNet with multiple SCs.

1.  Macro-cell Networks: A macro-cell network can supply extensive coverage by utilizing a high-power BS, which is usually utilized in cellular networks. The macro-cell network characteristics include: (i) being permanently located in a high area, such as skyscrapers or summits of mountains which can provide a line of sight over the neighboring buildings and obstructions; (ii) having a high transmission space and a massive coverage region, where the radius of the cell ranges from 1 to 25 km. Moreover, the space between adjacent MBSs is large; (iii) shadowing, fading, and interference of multipath have a significant impact on the cell-edge user QoS; and (iv) due to the existence of uncovered or hot areas because of unevenly distributed serving demands, the indoor users' QoS is much lower when serviced by the MBS [54].
2.  Microcell Networks: A low-power BS is used to serve the micro-cell network that is always established in highly populated metropolitan areas, such as shopping malls [55]. This network's coverage radius ranges from 200 m to 1 km, which is significantly less than that of the macro-cell network. Meanwhile, with low-power BSs, the frequency reuse distance decreases, while the number of channels and the density of traffic both increase substantially [56].
3.  Pico-cell Networks: A pico-cell network spans a significantly lower area (between 100 m and 200 m) when compared to a micro-cell network, such as training buildings. Typically, pico-cells are utilized to increase the coverage of indoor regions. As a result, they have the potential to minimize the uncovered areas of indoor communications [57].
4.  Femto-cell Networks: A femto-cell network (also known as a Home e-Node B) is a network with a small and low-power BS that is formed to increase the quality of communication in a home or small company. Using the home BS improves the QoS for indoor users [58]. Furthermore, femto-cells are significantly easier and more cost-effective to deploy than other types of cells. Besides that, femto-cells can be used to fill in the gaps between pico-cells and prevent the loss of signal via buildings. The

fundamental distinction between femto-cells and pico-cells is that the users' number in femto-cells is less than in pico-cells [59].

Based on the above explanation, the properties of various networks are briefly described in Table 1. The term "radius" refers to the BS's transmission radius. While the term "power" refers to the BS's maximum transmitting power. Moreover, the term "scenario" represents the application context in which each cell operates.

**Table 1.** A comparison of several network types [60].

| Cell | Scenario | Power | Radius |
|---|---|---|---|
| Femto | Home, small enterprises | [0.01, 0.2] | [0.01, 0.05] |
| Pico | Office building, underground parking | [0.25, 2] | [0.1, 0.2] |
| Micro | Shopping malls, railway station | [2, 20] | [0.2, 1] |
| Macro | Mountaintop | [20, 160] | [1, 25] |

### 4.1.3. Interference in HetNet

The combination of such SCs grants offloading traffic from macro-cell and improves the experience of the network by associating UEs in SCs with minimum power transmission. However, this combination leads to significant ICI in networks, particularly for SC users at the cell edge. In general, macro-cells are often deployed in a cellular network by a reasonable network plan, whereas low-power small-cells are typically placed by the identification of coverage problems and traffic intensities (e.g., hotspots) in the network [61]. Various types of distribution scenarios are already available for HetNets. In a multi-carrier distribution, SCs use more various carrier frequencies than macro-cells. This method efficiently minimizes ICI but does not guarantee optimum spectrum utilization [62]. On the other hand, the co-channel distribution is used by utilizing the same carrier for macro-cell and SC, in which the spectrum efficiency is optimized through spatial reuse and a prominent distribution technique in HetNets. Even though the co-channel technique enables excellent spectral utilization, it results in great ICI among macro- and small-cells [63].

Because of the synchronous operating of many SCs within these cells, different types of smart devices or small equipment are connected in an MBS in the environment of HetNet, resulting in co-tier interference, which is the interference between entities belonging to the same network or tier. In the case of a femto-cell network, the co-tier interference happens between nearby femto-cells. While the interference between entities belonging to diverse networks or tiers is referred to as cross-tier interference. Figure 6 depicts such interference between femto–macro and macro–femto networks [64,65]. These interferences are especially common at big gatherings when numerous users demand high throughputs, such as heavy data applications, internet browsing, and downloading/uploading images and videos. Accordingly, the ICI management and minimization approach would be created for next-generation cellular communication. Furthermore, all other interferences must be canceled to provide user fairness and QoS in wireless cellular networks [66,67].

### 4.1.4. Related Works in HetNet

Depending on the cell types and distinct interference discussed above, we will present the most recent solutions conducted to mitigate co-tier interference ([68–74]), cross-tier interference ([75–78]), or hybrid interference ([79,80]) as follows:

a.     Co-tier Interference Solutions

In [68], the authors investigated the interference management for both uniform small base station deployment (U-SBSD) and non-uniform small base station deployment (NU-SBSD) in a two-tier uplink HetNets scenario. A non-uniform SBS deployment (NU-SBSD) with fractional power control (FPC) and reverse frequency allocation (RFA) in the MBS coverage area was proposed to mitigate uplink ICI and improve uplink coverage performance. Simulation results stated that the proposed method enhances the edge users' coverage and ICI significantly by NU-SBSD with FPC and RFA as compared with U-SBSD.

The improvement of maximum uplink coverage was around 39.8% in the macro-outer region by employing NU-SBSD with FPC and RFA, compared with U-SBSD. Similarly, the improvement of maximum uplink coverage was about 18.34% in the macro center region by employing NU-SBSD with FPC and RFA. However, increasing the value of the fractional path loss compensation factor results in a reduction in uplink coverage probability because of high path loss and interference, which causes a degradation in the SINR of the proposed system.

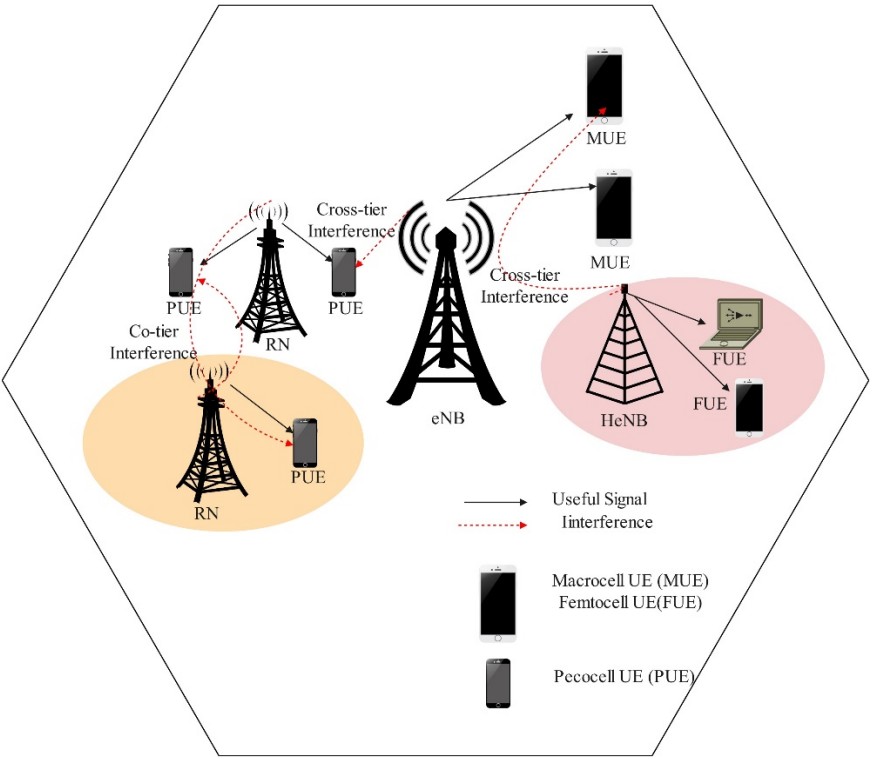

**Figure 6.** The interference in HetNet.

In [69], the authors proposed a distributed joint interference management (DJIM) algorithm for various domains in downlink heterogeneous ultra-dense small-cell networks (UDSNs) to mitigate the interference and maximize the total network's throughput. The frequency-domain orthogonal frequency division multiple access (OFDMA) scheduling was implemented to assign the appropriate sub-channels and construct reliable transmission links to mitigate adjacent co-tier interference in a single slot. The proposed algorithm enables each small-cell base station (SCBS) to self-organize and interact into a stable overlapping coalition structure, eventually reducing multi-domain interference and attaining an optimal cost-benefit tradeoff. According to the simulation results, the proposed algorithm delivers a significant improvement in overall performance in terms of total throughput. Nonetheless, the impact of the ICI that minimizes the system's throughput was not considered in this model.

In [70], the authors investigated the interference management in heterogeneous small-cell networks in uplink transmissions where interference cancellation strategies based on two sophisticated waveforms universal filtered multi-carrier (UFMC) are proposed to reduce both inter- and intra-cell interference in heterogeneous small-cell networks (HetSC-Nets). The first proposed strategy was the dynamic spectrum allocation strategy based on UFMC, and the second strategy was the precoding strategy based on UFMC. The proposed precoding strategy based on UFMC outperformed the dynamic spectrum allocation strategy based on UFMC in terms of BER performance. Additionally, the precoding strategy based on UFMC achieved a greater data rate than the dynamic spectrum allocation strategy based

on UFMC. Nevertheless, the non-uniform distribution for both users and BS that affects the system's power consumption was not considered in these strategies.

In [71], the authors investigated the interference management in two-tier downlink HetNets with hotspot centers, where each tier's BSs have varying transmitting powers, deployment densities, and connection reliability. The MBSs' hotspot centers are dispersed uniformly by two homogeneous PPPs. Several distance distributions were derived, including a joint distance distribution between a typical user and cooperative open-access mm-wave SBS, as well as a distance distribution between a typical user and non-cooperative open-access mm-wave SBS. Therefore, expressions for several performance metrics were obtained, which include association probability, SINR coverage probability, and ergodic capacity, under these situations. The numerical result indicated the best ratios of standard deviation for these performance indicators, as well as insights into the deployment of future networks. However, this study did not take into consideration the association probability for a variable number of SBS in each cluster that affects the ergodic capacity and data rate of the proposed system.

In [72], the stochastic geometry-based PPP approach was proposed to alleviate the relay-user-interference (RUI) and inter-relay-interference (IRI) between the relay and user link and then enhance the capacity of the system and the data rate of users. The proposed approach considered a multi-hope communication network for the mm-wave frequency band. As well, a full-duplex (FD) relaying model was considered and the locations of the source, relays, and users were modeled as point processes in the spatial domain. Furthermore, the Rayleigh fading channel with NLOS transmission at 28 GHz frequency was considered. The simulation results stated that the proposed scheme not only minimizes the probability of outage but also enhances the probability of success and ergodic capacity of the user, as compared with the traditional multiple antenna UDN models. It was also stated that deploying more Multiple-Input Multiple-Output (MIMO) antenna configurations can boost the probability of success by 6% and the ergodic capacity of the system by 400%. Yet, the users' mobility that increases the power consumption of the proposed approach was not considered in this study.

In [73], a novel interference minimization and radio RA management scheme for victim femto-cells (VFCs) was proposed to increase user throughput while decreasing co-tier interference for ultra-dense femtocell networks (UDFNs). The proposed semi-clustering of the victim-cell (SCVC) approach focuses mainly on the status of users either critical or non-critical to categorize VFCs and their aggressors. The proposed scheme was compared with the femtocell cluster-based resource allocation (FCRA) scheme. The simulation results stated that the proposed scheme outperforms the FCRA scheme in terms of critical user mean throughput, victim femtocell capacity, and resource usage percentage by around 185%, 64%, and 31%, respectively. However, the power spectrum efficiency that mitigates the co-tier interference was neglected in this study. The author suggested that SCVC technology can be practiced and evaluated in the future to determine its usefulness for eliminating co-tier interference and optimizing radio resource usage in the next 5G ultra-dense networks.

In [74], the authors proposed a multi-metric clustering with differential interference alignment (MMC-DIA) technique to take advantage of small-cell users' performance in a heterogeneous communication environment. The proposed technique operates in two main phases namely clustering formation and differential interference alignment. In the clustering process, sum-rate optimization objective-based clustering of small-cell users is used to maintain communication efficiency. For the comparative analysis, the proposed technique could maximize the SE and sum rate by 6.84% and 11.18%, respectively, along with DoF, regardless of the varying size and transmitting power. Similarly, for the varying power transmission, the proposed technique achieved 5.85% and 6.292% better SE and sum rates, respectively. Nonetheless, the dynamic heterogeneous environment to measure and address the influence of time events and interference that have a large impact on the power consumption for the proposed technique was not considered in this technique.

b.    Cross-tier Interference Solutions

In [75], the authors investigate the interference management in two-tier uplink and downlink HetNet scenarios where the uniform and non-uniform SBS deployment were considered. A soft frequency reuse (SFR) scheme combined with a power control factor was proposed. The proposed scheme alleviates the significant interference received from MBS because of offloaded users with optimum resource utilization. The SBSs were distributed uniformly using the PPP, while Rayleigh fading with standard path loss model and Addition Wight Gaussian Noise (AWGN) was considered. Numerical results stated that the suggested model achieves a greater probability of coverage because of the reduced interference and efficient use of SBS resources. Moreover, the results stated that the optimum radius for MBS's interior coverage region is around 70% of the radius for MBS's edge coverage region. In addition, the coverage area was improved when the SINR value decreased because of the increase in the number of users. However, it was found that when the MBS and SBS densities increase, the interference also increases, which minimizes the probability of coverage of the proposed system.

In [76], a new SFR algorithm was proposed to mitigate interference and maximize the network's throughput. The proposed scheme involves switching on/off SCs based on their interference contribution rate (ICR) values. SFR divides each SC into two zones: the center zone and the edge zone. The proposed scheme beats the previous switching on/off systems in terms of total system data rate, normalized traffic losses, power efficiency, and outage probability. Moreover, the results revealed that both irregular and circular forms have satisfactory performance in the center zone. Thus, the circular center zone was recommended for use with omnidirectional antennas due to the ease of deployment. However, the optimum proportion relative to the SC radius is 50%. Nevertheless, the end-to-end delay which affects the reliability of the proposed system is neglected in this study.

In [77], a multi-level SFR scheme for downlink HetNet was proposed to enhance the cell throughput and area spectral efficiency (ASE) while reducing the probability of outage due to ICI. In the proposed scheme, a mutually exclusive spectrum (MES) is associated with MC, SC, and edge users among different cells in the reuse system. The simulation results stated that the proposed scheme achieves significant improvement in cell throughput by 3.5-fold, ASE, and the probability of outage reduces by 5-fold compared with the traditional SFR scheme. Yet, the power allocation strategy which affects the average power efficiency of the proposed scheme was not considered in this work.

In [78], decupling association (DeCA) with reverse frequency allocation (RFA) in non-uniform HetNets (NUHs) based on the Poisson hole Process (PHP) for small base station deployment was proposed. The suggested technique aims to enhance uplink coverage due to improving the uplink signal-to-interference ratio (SIR) by mitigating uplink interference (UI), macro base station interference (MBSI), and ICI. The simulation results stated that the DeCA outperforms CA in terms of uplink coverage performance. In addition, the proposed system improvement was 76% in uplink coverage for SIR threshold values greater than 0 dB. Moreover, the high-value FPL compensation factor results in better coverage performance for the proposed system. However, the authors did not consider the user's mobility, which has a massive effect on the power consumption of the proposed system.

c.    Hybrid Interference Solutions

In [79], the authors proposed a novel interference mitigation and power allocation technique for downlinking with the MIMO technique in HetNet. The proposed technique, called Power Allocation-Based Interference Alignment and Coordinating Beamforming (PA-IA-CB), consists of two phases. The first phase consists of two steps of IA-CB, the first step constructs the transmit and receive beamforming vectors of Sus and SBSs to cancel inter-cluster and co-tier interference among SCs. The second step involves constructing transmit and receive beamforming vectors at macro users (Mus) and MBS, to eliminate inter-cluster interference within the MC. On the other hand, the cross-tier interference between the MC and the SCs is handled by the second phase. In this phase, that cross-tier interference can be

eliminated by adjusting the amount of power allocated to the MBS and SBSs and selecting SBS frequency resources that are different from those allotted to MBS. Simulation results stated that the proposed technique can be superior to the traditional MIMO-orthogonal multiple access (OMA) and MIMO-non-orthogonal multiple access (NOMA)-based HetNet in terms of overall system sum rate and outage probability at various SNRs levels and the ranges of coverage distance. Additionally, the results indicated that the proposed technique has the advantage of decreasing the signaling overhead because of the channel state information (CSI) sharing among SCs and MC. However, when the SNR value was extremely high, the system sum rate of the suggested technique decreased. This is because the impact of residue cross-tier interference becomes prominent in comparison with the noise level, which minimizes the sum rate of the proposed technique.

In [80], the model based on game theory which includes dynamic channel allocation, and a self-power optimization control method was proposed to address access exposure depending on priority by utilizing the idea of primary and secondary users. According to the simulation results, the suggested scheme was able to maximize the SINR level, channel usage, and system throughput capacity, as well as minimize outage probability, loopholes, and interference. Additionally, the proposed scheme assures high income for the operators while guaranteeing fair service costs for consumers. Nonetheless, the mobility of indoor and outdoor Ues that affect the system's power consumption was not considered in this model.

Table 2 illustrates the summary of the related works of HetNets discussed in this section.

**Table 2.** The summary of related works of HetNets in the literature.

| Issue | Methodologies | Advantages | Limitations/Future Work | Ref. |
|---|---|---|---|---|
| Mitigate uplink ICI and improve uplink coverage performance. | Non-uniform SBS deployments (NU-SBSD) with fractional power control (FPC) and reverse frequency allocation (RFA) in the MBS coverage area. | Enhance the edge users' coverage and ICI significantly by NU-SBSD with FPC and RFA as compared with U-SBSD. | Increasing the value of the fractional path loss compensation factor resulted in a reduction in uplink coverage probability because of high path loss and interference, which caused a degradation in the SINR of the proposed system. | [68] |
| Mitigate the interference and maximize the total network's throughput. | Distributed parallel iterative water-filling algorithm. | Deliver a significant improvement in overall performance in terms of total throughput. | The impact of the ICI was not considered. | [69] |
| Mitigate both intra-cell interference and inter-cell interference in HetSCNets. | Interference cancellation strategies based on two sophisticated waveforms universal filtered multi-carrier (UFMC). | Decrease the impact of frequency offsets and interference. | Both uniform and non-uniform distribution scenarios can be investigated. | [70] |
| Increase the network's capacity and range by reducing co-tier interference in downlink HetNets. | Fractional frequency reuse (FFR) and coordinated multi-point transmission (CoMP). | Give the best ratios of standard deviation for the precise coverage and ergodic capacity. | This study did not take into consideration the association probability for a variable number of SBS in each cluster that affects the ergodic capacity and data rate of the proposed system. | [71] |
| Alleviate RUI and IRI between the relay and user link to enhance the capacity of the system. | The stochastic geometry-based PPP approach. | Enhance the probability of success and ergodic capacity of the user by deploying more MIMO antenna configurations. | The users' mobility that increases the power consumption of the proposed approach was not considered in this study. | [72] |

**Table 2.** *Cont.*

| Issue | Methodologies | Advantages | Limitations/Future Work | Ref. |
|---|---|---|---|---|
| Increase user throughput while decreasing co-tier interference for UDFNs. | Semi-clustering of victim-cell (SCVC) approach. | Enhance the critical user mean throughput, victim femtocell capacity, and resource usage percentage by around 185%, 64%, and 31%, respectively. | The power spectrum efficiency that mitigates the co-tier interference was neglected in this study. | [73] |
| Improve the performance of SC by maximizing sum-rate and SE. | Multi-metric clustering with differential interference alignment (MMC-DIA) technique. | Maximize the SE and sum rate by 6.84% and 11.18%, respectively, along with DoF regardless of the varying size and transmit power. | The dynamic heterogeneous environment to measure and address the influence of time events was not considered. | [74] |
| Enhance user's SINR and alleviate the effect of interference because of user offloading in two-tier HetNets. | Conjoining an SFR scheme with a scenario of non-uniform SBS distribution, while considering the coverage probabilities for both uniform and non-uniform distribution scenarios. | -Maximize the probability of coverage because of reduced interference and efficient use of SBS resources. -Improve the coverage area when the SINR value decreased because of an increase in the number of users associated. | When the MBS and SBS densities increased, the interference also increased, which minimized the probability of coverage of the proposed system. | [75] |
| Mitigate the interference and enhance the power efficiency in 5G HetNet. | A new SFR algorithm based on their ICR values. | Maximize the total system data rate and power efficiency while minimizing the normalized traffic losses and outage probability. | The end-to-end delay which affects the reliability of the proposed system was neglected in this study. | [76] |
| Enhance the cell throughput and ASE while reducing the outage probability by minimizing inter-cell interference. | Multi-level SFR scheme. | Achieve significant improvement in cell throughput and ASE and reduce the outage probability. | The power allocation strategy, which affects the average power efficiency of the proposed scheme was not considered. | [77] |
| Enhance uplink coverage by mitigating uplink interference (UI) for MBSI | Decupling association (DeCA) with reverse frequency allocation (RFA). | Improvement of the uplink coverage performance by 76% for the SIR threshold value greater than 0 dB. | The authors did not consider the user's mobility, which has a massive effect on the power consumption of the proposed system. | [78] |
| Increase system sum rate by eliminating the inter-cluster and co-tier interference | Novel Power Allocation Based Interference Alignment and Coordinating Beamforming (PA-IA-CB). | Increase the overall system sum-rate and outage probability at various SNRs levels and the ranges of coverage distance. | When the SNR value was extremely high, the system sum rate of the suggested technique decreased. | [79] |
| Maximize QoS while reducing interference and increasing capacity in HetNets. | The advanced hybrid access approach in conjunction with the game theory includes dynamic channel allocation and a self-power optimization control method. | -Maximize the SINR level, channel usage, and system's throughput capacity. -Minimize outage probability, loopholes, and interference. | The mobility of indoor and outdoor Ues that affect the system's power consumption was not considered in this model. | [80] |

### 4.2. Device-to-Device (D2D)

The persistent demand for a maximum data rate with reduced end-to-end delay is one of the most significant defiance facing telecommunication providers. One method to accomplish it is via D2D communication. D2D communication allows nearby devices to communicate between them directly without passing into the BS. Due to the very low

latency associated with D2D communication, it has garnered considerable attention from researchers operating on promising B5G cellular communication networks [81,82]. D2D communication can use both licensed and unlicensed cellular spectrum which is referred to as in-band and out-band communication, respectively. In the first situation, both D2D and cellular communication can coexist on the same licensed cellular spectral; this is referred to as the underlay mode of D2D communication. However, it causes increased interference between cellular and D2D users. To address this issue, a novel communication mode called overlay D2D communication was suggested, which allows D2D users to utilize a portion of cellular resources that are not allotted to typical cellular users. To avoid spectrum waste, the effectiveness of recourse allocation must be considered in the overlay mode. Concerning D2D communication performance, it provides more advantages in comparison with traditional cellular communication, when it is viable technically. The D2D communication technique is characterized by transparency besides being extremely effective in terms of massive spectrum efficiency, low energy consumption, and low latency. Thus, local traffic management becomes easier for user Ues that communicate directly in a certain vicinity. Another advantage of D2D communication is that it allows for computational offloading. D2D users under the environment of a static network can utilize D2D links for offloading computationally intensive activities to adjacent D2D users [83]. The mechanism of mode selection in D2D communication enables devices to simply move from the infrastructure communication path to the direct communication path. This helps to decrease network congestion. Economically, D2D communication has a significant role to play in commercial, e-commercial, and social apps, among others, where users can immediately share important information locally [84,85]. Figure 7 shows the typical D2D communication scenario [86].

### 4.2.1. Unique Features of D2D

1. Single-hop communication: A single hop is required for communication between the devices. Communication in D2Drequires fewer resources, resulting in the effective use of the spectral. Because proximity users connect directly with one another in D2D communication, latency is significantly decreased. These D2D communication features also assist the operators of mobile networks [87].
2. Reusability of the frequency: When D2D communication is used in cellular networks, the same frequency is shared by both D2D and cellular users. This enhances frequency reuse, hence optimizing the frequency reuse ratio [88].
3. Power levels optimization: The existence of D2D links between close-by devices results in minimum transmission power over a short distance. This extends the device's battery life. As a result, D2D communication in cellular networks can achieve improved energy efficiency (EE) [89,90].
4. Increased area of coverage: Since D2D communication is feasible via relays, this allows for communication over larger distances, thus expanding the entire coverage area.

### 4.2.2. Communication Scenarios of D2D

There are new features defined in 3GPP Release 12 which enable the use of eNBs and core networks to facilitate D2D communication. Figure 8 shows the D2D communication scenarios in 3GPP Release 12. Three coverage situations are specifically addressed, as follows:

1. In coverage mode: In this communication mode, all Ues are within the eNB's coverage.
2. Out of coverage mode: In this communication mode, none of the Ues are under the eNB's coverage.
3. Partial coverage mode: In this communication mode, certain Ues are covered by the eNB while others are not. Ues under the eNB's coverage communicate with Ues that are not within the eNB's coverage [91].

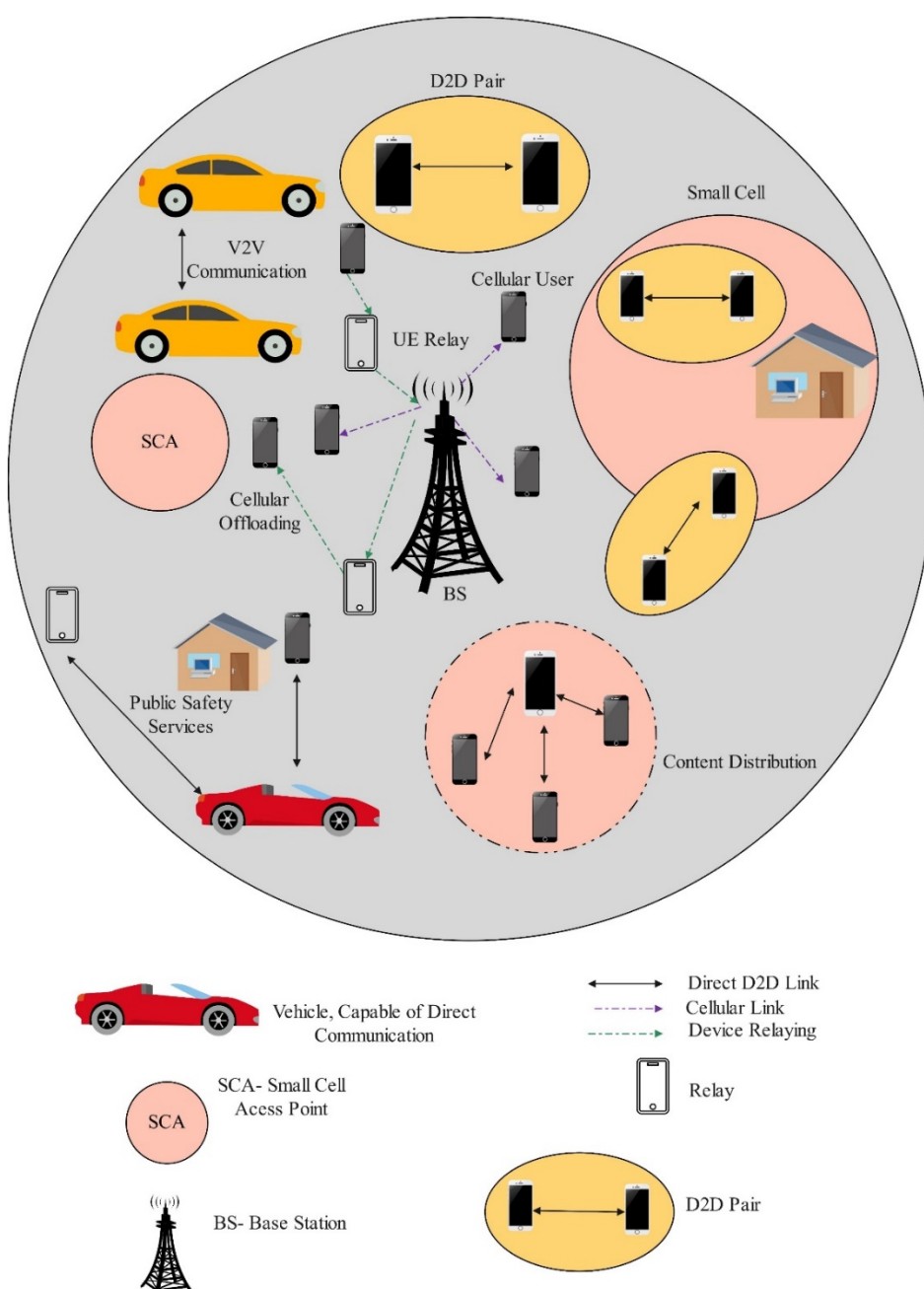

**Figure 7.** The General Scenario supporting D2D communication.

### 4.2.3. Interference in D2D

Interference management represents one of the major significant defiances for D2D communication. As explained earlier, the participating mode is the preferred mode for operators to maximize spectrum efficiency. However, this results in an interference problem. Because many cellular and D2D users utilize the same spectrum portion, they might cause interferences with each other.

To accommodate D2D communication, the cellular network's design was modified to contain two tiers instead of one [92,93]. The first tier is the traditional macro-cell tier, in which the BS and device communicate with each other. The new tier, known as the device tier, encompasses D2D communication. As a result, this type of system is referred to as the construction of a two-tier or cellular system. The device tier is an unregulated and arbitrary distribution of D2D user equipment (DUE). The new construction can significantly enhance data rate, probability of coverage, and end-to-end delay if constructed

accurately [94]. However, it offers many technical defiances and problems for both device and cellular user equipment. Due to these defiances, one of the most crucial concerns for D2D communication in participating mode is interference management between cellular and D2D user equipments, in which the same frequency resources are utilized for both D2D and cellular communication. To maximize spectrum efficiency, it is preferable to use D2D communication in a participating mode. However, this creates significant interference management defiances since, in comparison to the scenarios of cellular communication, the system must manage new interference conditions. The total capacity and spectrum efficiency of the cellular system deteriorate if the produced interference is not adequately controlled, which would reduce the possible advantages of D2D communication. The most prominent interferences seen in D2D communication are classified into two categories: network domain (co-tier) and frequency domain (cross-tier) [95].

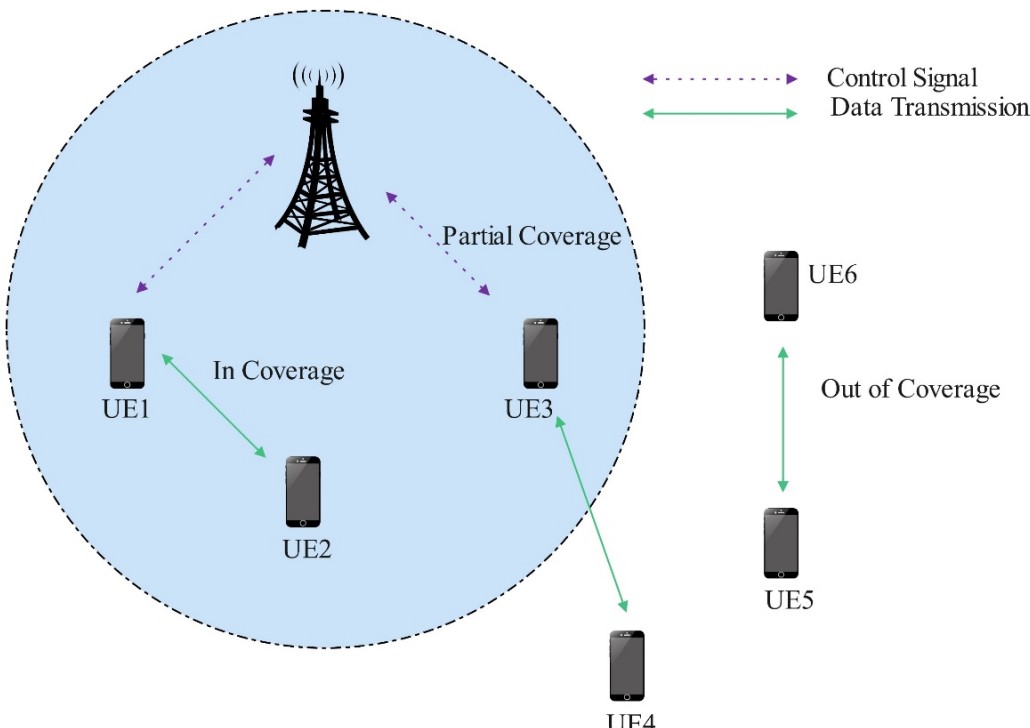

**Figure 8.** The scenarios of D2D communication supported in 3GPP Release 12.

The co-tier interference between D2D users happens when one D2D user communicates with another D2D user in the same tier. To establish a direct connection between D2D users, the SINR value should be greater than a preset threshold value. Otherwise, a direct connection link cannot be created if the SINR of DUE falls below the set threshold value due to co-tier interference. Co-tier interference occurs in OFDMA systems when the same resource block set is assigned to several DUEs. The D2D pairs which are allocated the same cellular frequencies are always subject to interference from the D2D transmitter to the D2D receiver, irrespective of the frequency reuse direction (Uplink (UL)/Downlink (DL)).

On the other hand, cross-tier interference occurs when network elements are from different tiers. Cross-tier interference can occur between (i) a cellular user equipment (CUE) and a DUE, or (ii) a CUE and many DUEs. This type of interference happens when a cellular user's assigned resource blocks are reutilized by one or more D2D users. In this form of interference, the aggressor (interference source) and the interference victim vary based on the direction of resource reuse (UL/DL) [96]. Figure 9 shows B5G HetNet D2D interferences. The prominent cross-tier interference in D2D communication is classified into two scenarios:

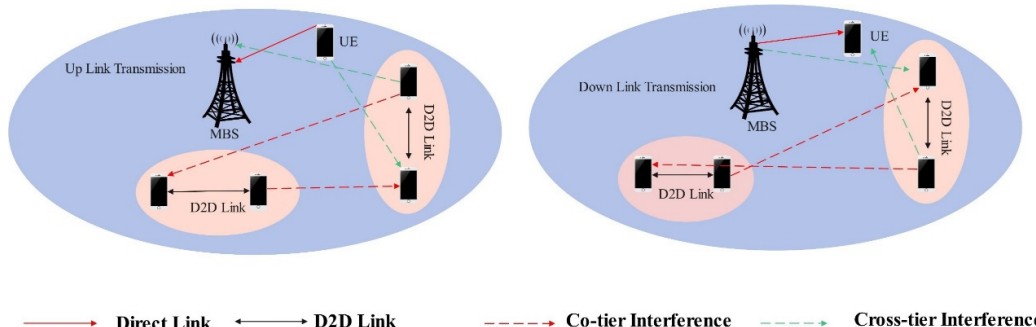

**Figure 9.** The interference in D2D HetNet.

　Scenario 1: D2D-cellular network interference: When the same frequencies of the uplink CUEs are reused by D2D users, the D2D transmitter interferes with the BS and the uplink cellular user interferes with the D2D receiver.

　Scenario 2: Cellular network-D2D user interference: When downlink frequencies from the licensed spectrum are reused by D2D communications, the BS interferes with D2D receivers, while the D2D transmitter interferes with the downlink cellular user.

　Finally, both co- and cross-tier interference from the D2D transmitter can be reduced at a D2D receiver using a suitable power allocation strategy, spectrum allocation strategy, or both.

### 4.2.4. Interference Control Level

　Generally, the strategies of interference management can be categorized as centralized, semi-distributed, and distributed according to the scenario operation. Figure 10 depicts the classification of the interference control level.

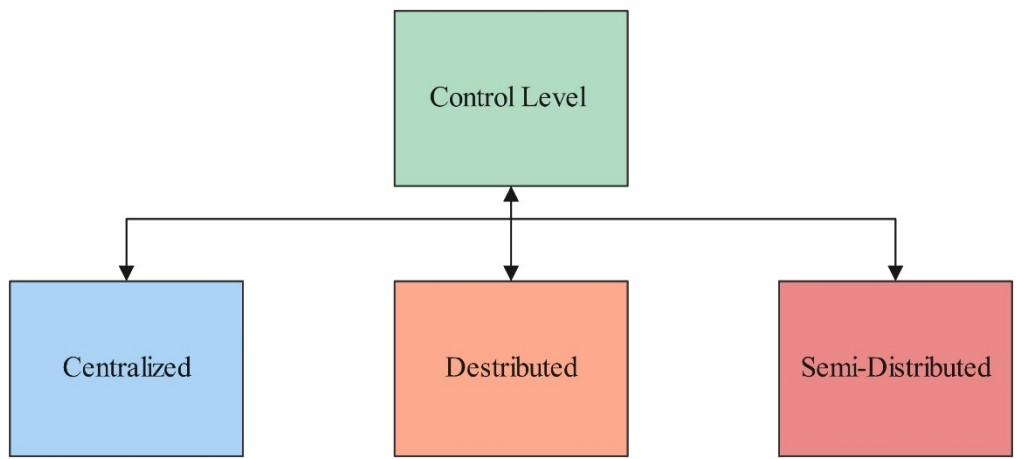

**Figure 10.** Control level of interference in D2D communication.

1.　Centralized

　In the centralized method, the interference between D2D and cellular users is completely managed by BS. This central entity combines information about each user in the network, such as the quality of the channel, CSI, and interference level. Moreover, it selects the channels that must be allocated to each user in the network with the appropriate format and power level. The central entity assigns the resources to each CUE or DUE depending on the collected information. The major issue of centralized methods is the massive amount of signaling necessary to exchange CSI and feedback. Furthermore, because the process is conducted by a single entity, which must handle massive amounts of data, the complexity of interference management increases significantly with the users' number in the network. Therefore, centralized methods are appropriate only for limited-scale D2D networks.

2. Distributed

In a distributed method, the interference management process does not need a central entity and is conducted independently by DUEs. Due to finite CSI and feedback, the distributed method minimizes control and computational cost. Thus, due to the difficulty of interference coordination, this method is better suited for large-scale D2D networks.

3. Semi-distributed

Although both centralized and distributed methods have benefits and drawbacks, trade-offs can be made between them. Interference management strategies of this type are referred to as semi-distributed or hybrid. Various levels of participation can be established in the strategies of semi-distributed interference management. Such strategies can be appropriate for relatively massive networks [88,97].

### 4.2.5. Related Work in D2D

As previously stated, power allocation strategies, spectrum allocation strategies, and hybrid strategies are used to mitigate co- and cross-tier interference in D2D communication networks. In this subsection, we present the most recent studies conducted in D2D on power allocation strategies ([98–100]), spectrum allocation strategies ([2,101–107]), and hybrid strategies ([108,109]), as follows:

a. Power allocation strategies

In [98], the side-lobe interference reduced vertex coloring (SIRVC) algorithm-based resource allocation for concurrent mm-wave D2D transmission was proposed. The resource allocation decreases interference and enhances the throughput by scheduling all the flows in a minimum number of time slots. The proposed algorithm was compared with the conventional vertex coloring (VC) algorithm and the time division multiple access (TDMA) method under the same condition. It was noted that the proposed algorithm outperforms conventional TDMA and conventional VC algorithms. The simulation results stated that the throughput per time slot of the proposed algorithm improved significantly by around 12.5%. Moreover, the interference due to the side-lobe was found to have an evident bad effect on the performance of the network. Furthermore, the results indicated that the throughput per time slot increases if the side-lobe interference is controlled by choosing an appropriate threshold. Nonetheless, the impact of the ICI, which affects the system throughput, was not considered in this study.

In [99], the authors proposed a model integrating a Gaussian directional antenna with a two-ray channel, creating an mm-wave D2D network that minimizes transmission power according to device allocation and beamwidth selection. A distributed structure is used to associate these devices while also utilizing the particle swarm optimization (PSO) technique to determine the most effective beamwidth to transmit and/or receive. Simulation results stated that, in comparison to existing interference management scenarios that disregard devices' transmission power, the proposed scenario was able to efficiently minimize transmission power and interference as well as maximize the sum rate of the system resulting in better performance. Nevertheless, the different heights of transmitters that have a large influence on the D2D power optimization were not taken into consideration.

In [100], the interference management (IM) and RA problems of D2D communications underlying HetNet was investigated. To minimize the interference and dead-zone obstacles, the downlink/uplink decoupling user association (DL/UL-DUA) procedure was considered which quantifies its ability on IM and network-wide D2D performance improvements. UL fractional frequency reuse (FFR) schemes adaptively determine where the sub-band-bandwidths (SB-BWs) are located depending on the density of UE, the density of e-node-B (eNB), and the on/off switching frequencies of the SCs. The simulation results showed that the proposed scenario considerably minimizes the number of CUEs in an outage. The CBM gave an accurate performance to the comprehensive solution with significantly decreased running time. Nevertheless, the impact of intra-cell interference

was ignored in centralized systems since it was assumed that eNBs distribute RBs of an allocated SB orthogonally. This causes system throughput degradation.

b.     Spectrum allocation strategies

In [2], a new PPP technique that depends on stochastic geometry to model the SINR, ergodic capacity, outage probability, and probability of success for the D2D-enabled co-operative cellular network was implemented. The proposed technique included a wide variety of interference by considering a multiple-hope high-density D2D-enabled cooperative cellular network in which the signal is conveyed from the BS to D2D bypassing three network hops: BS to RN, an RN to a CU, and CU to D2D user (DU). In addition, the FD mode and decode-and-forward (DF) protocol were utilized by RNs. Finally, the performance of the proposed approach was compared with the grid model and traditional multi-antenna ultra-dense network (MA-UDN) approaches. The simulation results stated that the proposed approach outperforms the grid model and traditional multi-antenna ultra-dense network (MA-UDN) approaches. However, the transmission power factor for the BSs, RNs, Cus, and D2D users that has massive effects on power consumption was not considered in this study.

In [101], the authors proposed a distributed resource allocation based on a one-to-many matching algorithm. This algorithm aims to minimize the co-/cross-tier interference between D2D and cellular communication in HetNet while maximizing the network data rate and satisfying the QoS requirements of D2D communication. The proposed algorithm can be used to allocate the resources of D2D communication in both half-duplex and full-duplex modes with considering the SIC in full-duplex mode due to self-interference. Simulation results stated that the proposed algorithm can realize the performance of the network nearby 93.7% of optimum performance with less overhead and complexity. Moreover, the proposed algorithm converged to a stable matching and terminated after a finite number of iterations. Nonetheless, if the density of FBSs increases, co/cross-tier interferences on network communications also increase. This leads to a decrease in the number of communications that reuse sub-channels.

In [102], a novel overlapping coalition formation game (OCFG) was proposed to solve the issue of mutual interference management and allocate resources in D2D communications as well as, enhance the average rate of all the D2D links in uplink D2D communications underlying cellular network concerning the QoS of every CU and D2D links. In the D2D links reselection process, the singular reward of each D2D link in terms of data rate and the overall utility of the coalitional structure in terms of the average data rate of all coalitions were compared. The simulation results stated that the proposed algorithm can enhance the performance significantly as compared with other existing schemes. Yet, when the D2D links' transmission power increased, the interference between UEs also increased, and this caused a system throughput degradation. That means increasing the D2D links' transmission power cannot always optimize the system performance.

In [103], the D2D-eICIC algorithm was proposed to improve the SE performance of eICIC applied to HetNet. The proposed algorithm uses D2D communications to assist in delivering the downlink data transmission to macro user equipments (MUEs) during almost blank subframes (ABSs). The proposed algorithm's performance depends on the traffic load at the SBSs and the relay channel conditions. The simulation results stated that the SE and sum rate of the proposed algorithm achieves better performance than the traditional eICIC and the baseline schemes in moderate traffic load conditions. However, the mobility of devices that affect the system power consumption was not considered in this study.

In [104], a new architecture called NOMA-V2X was proposed for 5G-enabled vehicle networks to enhance the network's throughput. In NOMA-V2X architecture, there are three main kinds of communication groups that coexist: V2I groups, multi-V2V groups, and uni-V2V groups. In the proposed system, multi-V2V and downlink V2I communications are subjected to two types of interference concurrently, while the uni-V2V communication group is subjected to just inter-group interference. The simulation results stated that the

network throughput is significantly improved with the help of the proposed resource allocation protocol for the proposed system. Nonetheless, increasing the number of vehicles caused an increase in the communication groups assigned to the same RB, resulting in increased interference between the communication links. Moreover, when the transmission power of the roadside unit (RSU) was increased, the total data rate was minimized proportionally. This is because the level of interference becomes more severe as the transmit power of the RSU increases. Furthermore, the total data rate of the network decreases when the speed of the vehicle increases.

In [105], the authors proposed a decentralized interference management method to improve the overall signal-to-interference-plus-noise ratio (SINR) of the network system while reducing the complex computational load on MBS. The proposed Interference Management (IM) divides the interference into cross-cluster interference and intra-cluster interference and treats them separately. The performance analysis of the spectral clustering technique was compared with benchmark clustering techniques such as Kernel means (K-means) and Kernel Principal Component Analysis (KPCA) clustering. It was found that the proposed technique extremely reduced the average cross-cluster interference. Moreover, the proposed dynamic resource allocation scheme reduces the inter-cluster interference which results in maximizing the overall SINR of the network. However, when the RBs were reused among the D2D Ues set, the proposed system suffers from severe interference.

In [106], a novel RA for downlink communications underlying MIMO-NOMA cellular network was proposed for maximization of SE while guaranteeing the QoS of both CUEs and D2D pairs and providing interference protection for CUEs and receivers device user equipment (RDUE). To enhance network SE, an optimum power allocation approach depending on particle swarm optimization (PSO) was proposed for both CUEs and DUEs, while preserving CUEs from intra-cluster interference induced by TDUE and ensuring QoS for CUEs and D2D Pairs. Simulation results stated that the proposed RA algorithm provides massive spectrum and energy efficiency as compared to traditional D2D communications that use MIMO-OMA cellular networks. Yet, when the CUEs and D2D pairing numbers were increased with a decrease in the number of clusters, the SE significantly decreased for the proposed model.

In [107], the decentralized algorithm based on the auction approach was proposed to solve the RA issue in multi-tier HetNet and to realize maximum SE and total data rate without causing prominent cross-tier and co-tier interference to the macro user equipments (MUEs) and underlay user equipments (DUEs and SUEs), respectively. Furthermore, the Rayleigh fading, shadow fading, and path losses concerning distance were considered. The simulation results stated that the proposed approach has better performance than the optimum centralized RA algorithm. Moreover, the proposed approach achieved close to 80% of the maximum data rate with less overhead and complexity. Nonetheless, the selfish and bad-behaving transmitters that have large impacts on the data rate of the proposed approach were not taken into consideration.

c.    Hybrid strategies

In [108], the suboptimal DDT-DMU user grouping and RA scenarios were proposed for both DMGs and CMUs to optimize the total network's sum rate while preserving the SINR of the CMUs and DMUs. The DMGs in the underlying cellular network are constructed depending on SIC decoding to decrease intra-user interference. The proposed scenario was compared with the existing joint user clustering and power allocation (JUCPA) and joint spectrum and power allocation (JSPA) scenarios. Numerical results stated that the proposed scenario achieves a better sum rate than the existing NOMA and OFDMA scenarios. However, increasing the number of DMGs in the cell resulted in a decrease in the chance of identifying the optimal RB because of increased co-channel interference. Hence, the time taken for the algorithm implementation increased due to the increase in the cross and co-channel interference via each RB, leading to a decrease in the DMGs sum-rate because of increased co-channel interference. Furthermore, increasing the number of CMUs

in the cell reduced the DMG sum-rate because it captures a large portion of the frequency resources required to acquire the minimum data rate.

In [109], the authors investigated the joint resource and power allocation issue for cooperative D2D users (CDUs) which multiplex Cus in downlink cooperative D2D heterogeneous networks (CDHN). The RA issue contains allocating spectrum RBs and selecting an idle user to act as a relay to aid the D2D links communication, whereas the purpose of PA is to minimize inter-user interference and enhance the QoS of communication. The analytical formulas for the aggregate throughput were derived for the suggested scenario. The analytical expression demonstrated that choosing an appropriate IU as a relay for each DU, distributing spectrum RB, and managing power for each IU, DU, and CU are three main factors affecting the aggregate throughput of CDHN under the constraint of the Cus' throughput demands. The simulation results stated that the suggested algorithm outperforms competing algorithms for various system parameters. Nonetheless, when the transmission power of BS increases, the interference to Dus also increases, which led to a reduction in the total throughput of the suggested scenario.

The summary of previous studies of D2D is presented in Table 3.

**Table 3.** The summary of previous studies of D2D in the literature.

| Issue | Methodologies | Advantages | Limitations/Future Work | Ref. |
|---|---|---|---|---|
| Minimize the main- and side-lobe interference as well as enhance the throughput in the mm-wave D2D network. | Side-lobe Interference Reduced vertex coloring (SIRVC) algorithm-based resource allocation. | Improve the throughput per time slot significantly by around (12.5%). | The impact of the ICI, which affects the system throughput, was not considered. | [98] |
| Minimize transmission power according to device allocation and beamwidth selection. | A model integrating a Gaussian directional antenna with a two-way channel. | -Minimize power transmission and interference. -Maximize the sum rate of the system. | -The different heights of transmitters that have a large influence on the D2D power optimization were not taken into consideration. | [99] |
| Alleviate the interference and dead-zone problems for D2D-Enabled DL/UL Decoupled Het-Nets. | Decoupling user association (DUA) procedure by using UL fractional frequency reuse (FFR) scheme. | Minimize the number of CUEs in an outage. | The impact of intra-cell interference was ignored in centralized systems since it was assumed that eNBs distribute RBs of an allocated SB orthogonally. | [100] |
| Produce D2D cellular networks devoid of interference. | A new PPP technique depends on stochastic geometry. | Maximize SINR, ergodic capacity, and probability of success as well as minimize the outage probability for the D2D-enabled cooperative cellular network. | The transmission power factor for the BSs, RNs, Cus, and D2D users that has massive effects on power consumption was not considered in this study. | [2] |
| Minimize the co/cross-tier interference between D2D and cellular communication in HetNet while maximizing the network data rate | One-to-many matching algorithm. | Realize the performance of the network nearby (93.7%) at optimum performance with less overhead and complexity. | When the density of FBSs increased, co/cross-tier interferences on network communications also increased. This led to a decrease in the number of communications that reuse sub-channels. | [101] |

**Table 3.** *Cont.*

| Issue | Methodologies | Advantages | Limitations/Future Work | Ref. |
|---|---|---|---|---|
| Allocate the resources to uplink D2D communications as well as mitigate the mutual interference between different Ues. | An overlapping coalition formation game. | Enhance the average rate significantly of all D2D links in uplink D2D communications. | When the D2D links' transmission power increased, the interference between Ues also increased, and this caused a system throughput degradation. | [102] |
| Enhance the SE and sum-rate as well mitigate the ICI in downlink HetNets. | The D2D-eICIC algorithm. | Improve the performance of SE and sum rate. | The mobility of devices that affect the system power consumption was not considered in this study. | [103] |
| Enhance the spectrum efficiency in both downlink V2I and multi-V2V groups by mitigating inter-and intra-group interference. | A three-dimensional matching method for allocating resources based on weighted interference hypergraph (IHG-3DM). | Improve the network throughput. | Increasing the number of vehicles caused an increase in the communication groups assigned, resulted in increasing the interference between the communication links. | [104] |
| Improve the overall SINR of the network system by mitigating intra-cluster and cross-cluster interference. | Spectral clustering technique with modified kernel weights with Dynamic resource allocation scheme using graph coloring. | Minimize the average cross-cluster interference and reduce the inter-cluster interference which resulted in maximizing the overall SINR of the network. | When the RBs were reused among the D2D Ues set, the proposed system suffered from severe interference. | [105] |
| Maximize the SE while guaranteeing the QoS of both CUEs and D2D pairs by mitigating inter-and intra-cluster interference as well inter- and intra-beam interference. | A novel graph theory-based interference-aware user clustering. | Provide massive spectrum and energy efficiency. | When the CUEs and D2D pairings number were increased with a decrease in the number of clusters, the SE significantly decreased for the proposed model. | [106] |
| Enhance the SE as well as total data rate by mitigating co- and cross-tier interference | The decentralized algorithm is based on an auction approach. | Maximize the SE as well as the total data rate with less overhead and complexity. | The bad-behaving transmitters that have large impacts on the data rate were not taken into consideration. | [107] |
| Optimize the total network's sum rate while preserving the SINR of the CMUs and DMUs as well as minimize intra-user interference. | Suboptimal DDT-DMU user grouping and RA scenarios for both DMGs and CMUs. | Maximize the sum rate. | Increasing the number of DMGs in the cell resulted in a decrease in the chance of identifying the optimal RB because of increased co-channel interference. | [108] |
| Minimize inter-user interference, enhance the QoS of communication and increase aggregate network throughput. | A quantum coral reefs optimization (QCRO) algorithm. | Maximize total throughput. | When the transmission power of BS increased, the interference to Dus also increased, leading to a reduction in the total throughput of the suggested scenario. | [109] |

### 4.3. Ultra-Dense Networks (UDNs)

The fast evolution of wireless cellular networks forces us to pay close attention to developing technologies that are critical for network performance enhancement. With the introduction of modern 5G wireless communications, the network's complexity will be more difficult. UDNs are considered the basic core of B5G systems. Without losing generality, UDNs are the fundamental technology that fulfills the essential demands of massive traffic requirements to be met by 2022 and beyond. The UDN is a network

distribution technique that involves the addition of low-power nodes to the network to strengthen hotspots, eradicate blind spots, enhance the coverage of the network, enhance energy efficiency, and complete spectrum resources utilization [69,110]. There are two types of BSs in UDNs: fully functioning access nodes such as Pico and Femtocells and accessory functioning access nodes such as relays and remote radio heads (RRHs) [111,112]. Indoor femto-cells have three modes of operation: open, closed, and hybrid. In the open-access mode, all Ues have unrestricted or priority access to the AP, but in the closed-access mode, a group of Ues is designated for scheduled transmission. In hybrid mode, all Ues have restricted access to the serving node [113]. The spectrum reuse factor may be considerably optimized by densely deploying SCs, which enhances network capacity, particularly in densely populated areas. The convenient UDNs scenarios are apartments, stadiums, offices, campuses, subways, and residential areas. A wide range of services will be available for each scenario, including HD-TV, online gaming, virtual reality, augmented reality, live video streaming, video conferencing, cloud storage, high-definition image upload/download, intelligent home control, etc. [114,115]. UDN's fundamental structure is defined in terms of SC or density of the access point, where the number of serving access points (AP) is equal to or more than the users' number [116]. The general configuration of the suggested B5G UDNs is illustrated in Figure 11.

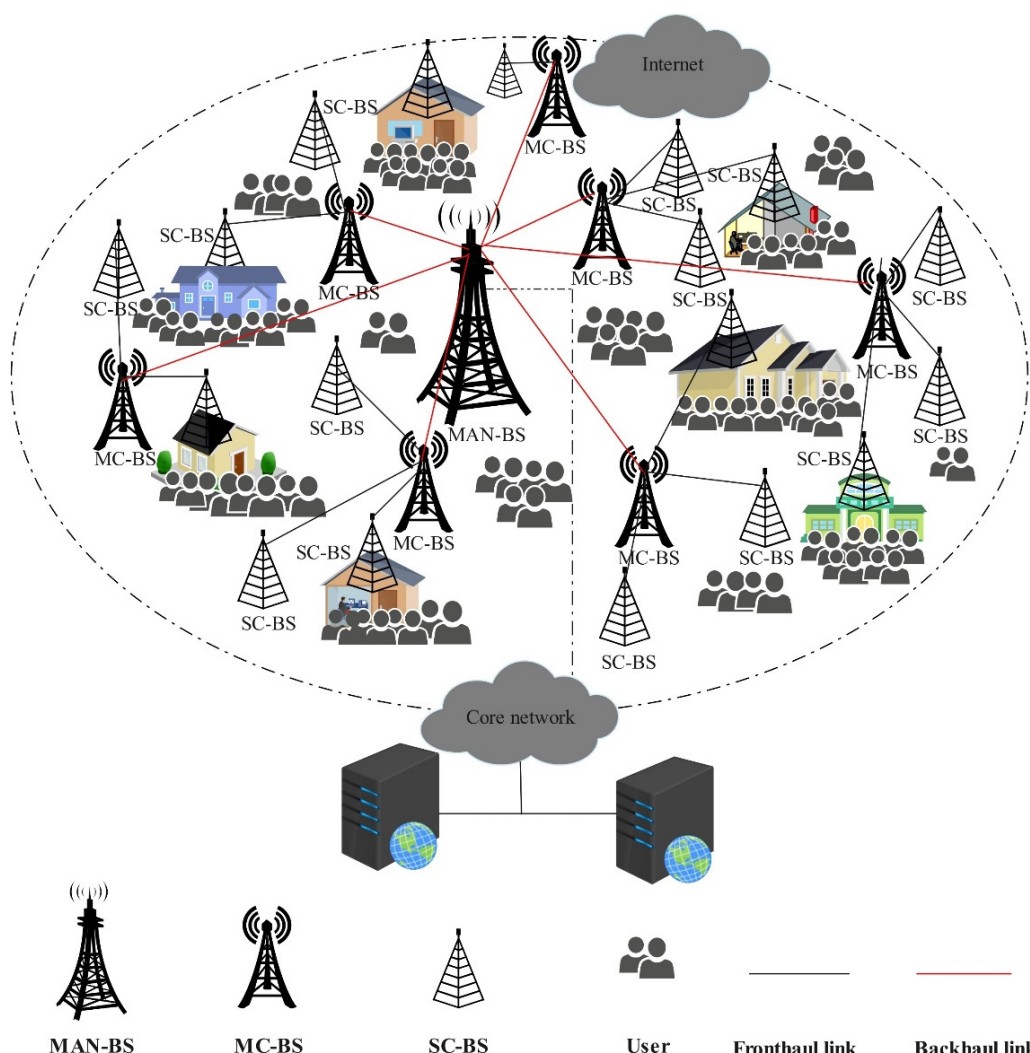

**Figure 11.** General structure of the proposed B5G UDNs.

### 4.3.1. Unique Features of UDNs

1.  A massive number of SCs and AP (more than or equal to the Ues number). The massive number of SCs can enhance frequency reuse in the same manner that adjacent distance and frequency reuse operate in macro-cells. The dense SCs increase the capacity of the network by offloading the traffic of macro-cell, balancing loads of the network, and minimizing congestion [117,118].
2.  Dense and extensively interconnected cross-tier distribution. This comprises macro-cell, SCs (femto-cell, pico-cell), relay nodes, D2D connections, etc., which boost the network environment's complexity. Due to the multi-tier distribution, the signals of various frequencies are sent throughout the overlapping region (e.g., macro-cell and SC). Furthermore, the proximity of SCs results in a great frequency reuse factor. Thus, the coordination of sophisticated interference is critical to reducing intra-tier interference and inter-tier interference, as well as assisting with resource management [119,120].
3.  Quick access and flexibility of switching (e.g., handovers). In the dense distribution scenario, the mobile UE may often swap the connection among access nodes, to get, the best service, optimal communications, and so on. The performance of high-quality handover (HQHO) is required to hand over smooth and seamless communications [121,122].

### 4.3.2. Interference in UDNs

Interference is a tricky problem in UDNs because dominating interferers occur near the intended receivers, as demonstrated in Figure 12. The coordination of interference is a sophisticated issue in UDNs because of the various BS density, which causes some BSs to interfere more than others. Fortunately, because of the relative number of Aps and Ues, several Aps may not have any linked Ues. Thus, shutting off such Aps, or minimizing their transmission power is a preferable strategy to minimize the impact of interference and total power consumption. For example, discussed three sleeping modes: cell-driven, core network-driven, and UE-driven [123,124]. The SC is triggered in the first mode if a planned active user is present. In the second mode, the network's central core has the authority to send a wake-up message to a specific BS. The last mode indicates that UE can wake up a neighboring cell by transmitting a wake-up message. The most popular types of interference in UDNs can be described as follows:

1.  Inter-Cell Interference: ICI occurs because of spectrum scarcity when the available spectrum is unable to meet the rising demand. To accommodate a rising number of Ues, frequency reuse mechanisms across various cells are developed. However, the ICI will be strict in UDN, as frequency reuse will be possibly increased by a factor of more than one, and will be more complex because of intensive deployment, near distance, irregular distribution, etc. Therefore, ICIC techniques should be improved to minimize ICI. The ICI can be minimized by the use of sophisticated receivers on the UE side, scheduling of joint cells on the network side, or joint collaboration between UE and the components of the network side [125].
2.  Multi-tier Interference: In UDN, both macro-cells and SCs are distributed through the network. Different emission powers, topologies of cells, radio access points [126,127], and other factors all contribute to the interference created by multi-tiers. For instance, SCs utilize the macro cell's frequency range, causing interference with the macro cell's UE (MUE), particularly the MUEs located at the cell edge (CE). At the CE, MUEs received a signal with significant fading and path loss [128]. When several SCs communicate over the same sub-channel, the interference with MUE will be more severe. Furthermore, because of the regulation of power, the MUE near the CE boosts its power emission, causing interference to the SC Ues [129].
3.  Small-to-small Interference (S2SI): Due to the high density of SCs and the topology of irregular distribution, the distributed method located on the SUE side, or the SC BS side is the preferable method to alleviate S2SI. The primary approaches for mitigating S2SI in SC BSs and SUEs are interference avoidance and interference elimination [130].

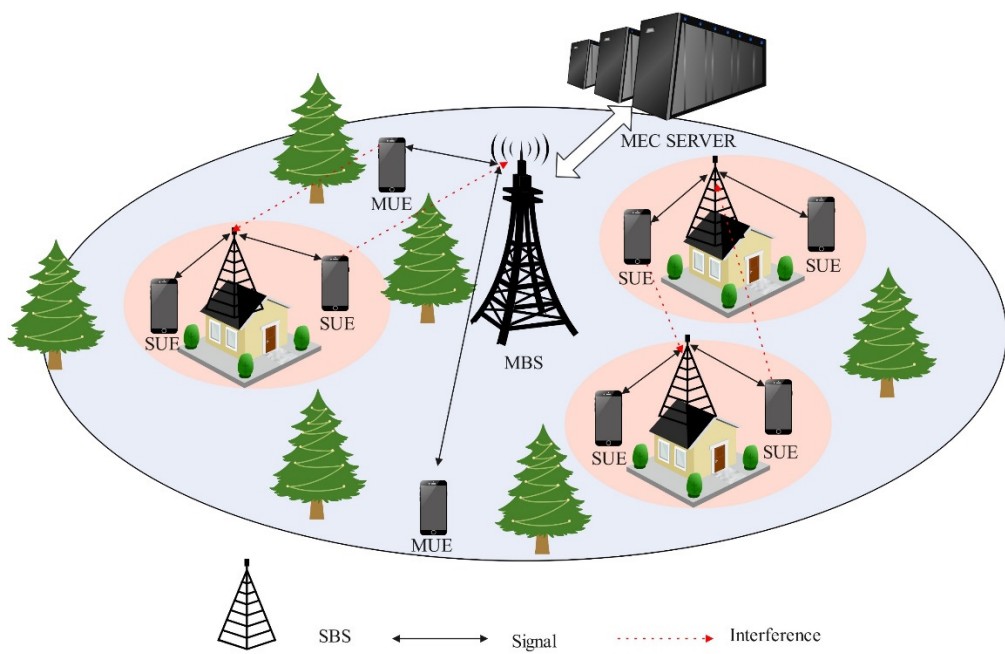

**Figure 12.** Interferences in UDNs Scenario.

Generally, to alleviate interference, current coordination systems are established concerning partitioning the resources depending on frequency-domain, time-domain, power domain, and spatial domain either on the UE side or network side, or a control combination between them [131]. On the other hand, overhead signaling is required for cell coordination. ICI can be alleviated by the cooperative macro-cell BSs and the assistance of UE. Specifically, the coordinating scheduling of the time domain, the signal orthogonalization of the frequency domain, and the coordination of spatial cells using advanced antennas can all be employed cooperatively to alleviate the ICI. The coordination of frequency cooperation and the sensing of the UE spectrum can be able to address the multi-tier interference. In the spatial domain, the interference can be mitigated by cell clustering cooperative combinations on the network side and advanced antenna of UE with interference cancelation. Due to the massive frequency reuse and near distance, S2SI is strikingly similar to ICI. Collaborative spectrum management, adaptive carrier selection, and adaptive power management are all possible strategies for mitigating S2SI. Interference can be controlled by the coordination of the network or through advanced receiver design [132].

### 4.3.3. Related Work in UDNs

Based on the above methods for reducing interference, in this subsection, we present the most recent studies conducted on the time-domain approaches ([133,134]), frequency-domain approaches ([135–141]), power-domain approaches ([142,143]), and spatial-domain approaches ([144–146]) in UAVs, as follows:

a.    Time-Domain approaches

In [133], the authors considered two-tier heterogeneous UDN (HUDN) with hexagonal macro-cells and PPP small-cell deployment. An interference-aware non-coherent coordinated multipoint transmission (IA-COMP) scenario was utilized to minimize both co-tier and cross-tier ICI for HUDN. As well, range expansion (RE) was used to optimize the load balance between macro-cells (MSs) and SCs in HUDN. The simulation results stated that the suggested technique can supply a more precise upper bound than the Monte Carlo simulation. Moreover, the system coverage increases with the larger one of the main ICI judging coefficients and the RE bias. However, the suggested technique resulted in difficult performance analysis because of the complexity of hexagonal networks.

In [134], the authors investigated the interference management problem for UDN in the TDD downlink scenario. A location-aware self-optimization (LASO) scheme was proposed for managing the downlink ICI in UDN as well as to improve the per-user throughput by adjusting downlink transmission power offset based on the effective provision of positioning. The simulation results confirmed that the proposed scheme achieves significant SINR gain and enhanced per-user-throughput, compared with the SC on/off-discovery signal (DS). Since the LASO scheme does not require DSs, it is not affected by the UE category and does not degrade network quality as a periodic interferer due to the DS transmission. Thus, the LASO scheme is a good solution for DL interference management in UDNs. Nonetheless, increasing the number of UEs increases the interference level, and this reduces the capacity of the system.

b.     Frequency-Domain approaches

In [135], the authors investigated the interference management in UDNs based on OFDMA in a two-tier downlink scenario. A centralized user-centric merge-and-split coalition formation game in which the users engage as players in the game was proposed to predict inter-user interference and leverage users' information (e.g., distance) to aid in the distribution and utilization of subchannels. The simulation results demonstrated that the proposed techniques eliminate intra-tier interference effectively and increase total throughput significantly through the TDMA in the coalition MIMO scenario. Nevertheless, allocating orthogonal sub-channels for all users cannot be realized because of the imperfection of the available sub-channels. One common sub-channel can be associated with several users, which causes a massive CCI and minimizes the total throughput for the proposed system.

In [136], the authors investigated the resource management in downlink multi-user-centric UDN with a massive number of lightweight access points (Aps) managed by a cloud-based intelligent transport system (ITS) for mitigating both frequency handover and ICI. The simulation results stated that the proposed scenario provides better performance in terms of RA fairness compared with the Nesterov successive convex approximation (Nesterov SCA) algorithm, multiplicative update (MU) algorithm, and centralized algorithm under the same conditions. Moreover, the proposed scenario was found to have a significant impact on theoretical and practical aspects of future V2X communication in UDN. Yet, in this scenario, the SINR decreased significantly with the increment of the vehicle hotspot size. This is due to the difficult management between VCs when increasing the multicast group size, which causes a decrease in the system data rate. In this study, the effect of intra-cell interference was not taken into consideration.

In [137], the authors investigated resource management in UDNs based on single-carrier OFDMA in an uplink scenario. A conflict-graph strategy based on machine learning that uses the uplink SINR and RB allocation data was proposed. Simulation results showed that the proposed strategy is both practical and precise. Therefore, this strategy was able to be implemented with network auto-adjustment and optimizing intelligent RA. Yet, the effect of CCI in this strategy was found to be severe due to the reuse of the RBs, resulting in throughput degradation.

In [138], the authors investigated a joint RA and SIC in UDNs based on NOMA in a two-tier downlink scenario. An interference management strategy was proposed that involves joint optimization of clustering, sub-channel allocation, and SIC. The simulation results indicated that the average capacity and spectrum efficiency for the proposed strategy increased significantly as compared with optimal Femto base station sub-channel allocation (OFBSSA) and cluster-based Femto base station sub-channel allocation (CFBSSA) strategies. However, the average capacity of the proposed system decreased significantly due to many reasons such as an increase the co-tier interference, the number of FBS users in overlapping areas, and interference among users.

In [139], the K-mean clustering algorithm was applied for determining the optimum number of clusters to increase network capacity while considering frequency reuse usage and inter-cluster interference. The BSs and UEs were distributed randomly throughout

the cluster using the PPP scenario. The simulation results indicated that the best number of clusters is around 13, and it was discovered that the operating frequency band has the greatest influence on the optimum number of clusters. However, when a certain threshold was crossed, the inter-cluster interference increased with the increment of the number of clusters. Nonetheless, when the number of clusters increases, the interference also increased, and this led to a decrease in the channel capacity of the proposed system.

In [140], the authors investigated the interference management problem for two-tier UDNs in the uplink scenario. A cross-tier cooperation load-adapting interference management (CCLA-IM) distributed strategy was suggested to minimize ICI by RA optimization between users in UDNs. The simulation results stated that the suggested strategy provides superior performance in terms of SE, SBSs throughput, EE, and ICI allocation whereas users' density and traffic loading were altered in UDNs. Nevertheless, the increased number of users who share the same bandwidth decreases the density of SBSs because of the increased uplink interference from the surrounding users. Furthermore, the increment number of users per SBS led to an increase in the mutual interference among various users served by various SBSs, which caused a decrease in the average EE of the proposed system.

In [141], the authors investigated the coordinative interference management in UD-SCN based on OFDMA in a two-tier downlink scenario. A new and simple-to-implement interference reduction technique that depends on a hierarchical clustering algorithm (HCA) between SBSs to calculate the member pairs was proposed. The simulation results indicated that the proposed scenario could increase the data rate of the network by 422.13 % for a network of 100 cells as compared with the non-cooperative scenario in a UD-SCN. Furthermore, it was especially suitable for hyper-dense deploying networks of SBSs. Yet, the effect of CCI in this technique was found to be severe due to the sub-channel being allocated to more than one SUE, resulting in data rate degradation in the proposed system.

c.　　Power-Domain approaches

In [142], a new non-cooperative game theory-based interference mitigation strategy for uplink power allocation was proposed to mitigate the ICI and optimize energy efficiency in the uplink mm-wave UDN multicarrier system. The simulation results demonstrated that the suggested strategy considerably improves EE performance while maintaining an acceptable SE performance as compared to previous iterative water-filling strategies. Moreover, the suggested strategy offered a low computational complexity. However, each SUE selected its own PA strategy depending on the assumption of optimizing its EE without considering the influence of other SUEs, which caused an increase in power consumption.

In [143], the authors investigated interference mitigation in downlink indoor coverage scenarios with autonomous UDN deployment. A completely distributed self-learning interference minimization (SLIM) scenario for independent networks under a model-free multi-agent reinforcement learning (MARL) structure was suggested for mitigating ICI, accommodating additional UEs, and decreasing the outage ratio of the system. The simulation results demonstrated that SLIM outperforms several existing known interference coordination schemes in mitigating interference and reducing power consumption while guaranteeing UEs' QoS for autonomous UDNs. Nonetheless, by increasing the number of users, the ICI increased as well, and the system became overloaded, leading to maximizing the outage ratio.

d.　　Spatial-Domain approaches

In [144], the authors considered a HetNet where a macro-cell layer provides essential service and coverage was overlayed by an ultra-dense layer of SBSs. A novel metric technique was proposed to optimize a UDN's downlink throughput with an appropriate degree of special spectrum reuse (SSR). The simulation results demonstrated that a UDN must achieve an optimal trade-off between reuse of the spectrum and interference to provide high throughput and low outage performance. Nevertheless, when the number of SCs increased, the outage threshold also increased. This is mainly due to the increased

users' number in the outage, specifically for the total reuse. This resulted in a decrease in the throughput.

In [145], the authors adopted chance constraint programming (CCP), in which occasional violations of the load threshold at BSs were permitted, and they introduced a control parameter, called risk level, to address traffic uncertainty while also achieving the trade-off between load balancing and the probability of constraint violation. The numerical results stated that the suggested strategy is resistant to traffic uncertainty. Furthermore, it was able to suppress severe interference and use the density of BSs to achieve superior load balancing performance compared to existing benchmark systems. However, a significant time delay could be observed for the proposed system if there was a high traffic level, in which each BS was compelled to offload the traffic from other BSs. This caused an increase in the congestion risk due to the arrival of burst traffic.

In [146], the authors investigated the small-cell clustering in UDN based on orthogonal frequency division multiplexing (OFDM) in a two-tier downlink scenario. A user-centric adaptive small-cell (SC) clustering strategy relying on an enhanced K-means algorithm was presented to decrease interference in UDN. The simulation results demonstrated that the proposed approach is capable of dynamically adjusting the number and size of SC clusters in response to the user's SINR and effectively reducing the complexity associated with the clustering process. It is important to mention that the radio RA strategy was not considered in this strategy, which has a significant effect on the optimal fairness among the users.

Table 4 summarizes the related works of UDNs discussed in this section.

**Table 4.** The summary of related works of UDNs in the literature.

| Issue | Methodologies | Advantages | Limitations/Future Work | Ref. |
|---|---|---|---|---|
| Minimize both the co- and cross-tier ICI and optimize the load balance between MSs and SCs in HUDN. | A coverage analysis technique based on MSG for an IA-COMP scenario | Supply a more precise upper bound and is hence more useful for practical scenarios. | The suggested technique resulted in difficult performance analysis because of the complexity of hexagonal networks. | [133] |
| Manage ICI and improve per-user throughput for UDN. | A location-aware self-optimization (LASO) scheme. | Significant SINR gain and enhanced per-user-throughput. | Increasing the number of UEs resulted in an increase the interference level, which caused a reducing the capacity of the system. | [134] |
| Increase total throughput, predict inter-user interference in two-tier downlink UDNs. | Centralized user-centric merge-and-split coalition formation game with Supplemental allocation algorithm (SAA). | Increase the total throughput significantly. | One common sub-channel can be associate with several users, which caused a massive CCI and minimized the total throughput for the proposed system. | [135] |
| Mitigate both frequency handover and ICI in dynamic virtual-cell-based UDNs. | Dynamic user-centric virtual cell (DUVC) scenario. | Better performance in terms of resource allocation fairness. | The SINR decreased significantly with the increment of the vehicle hotspot size. | [136] |
| To aid the implementation of optimal resource allocation and thereby reduce inter-user interference and co-channel interference. | Conflict graph strategy based on machine learning. | Network auto-adjustment and optimizing intelligent RA. | The effect of CCI in this strategy was found to be severe due to the reuse of the RBs, resulting in throughput degradation in the proposed scenario. | [137] |
| Minimize cross-tier interference and intra-cluster interference in two-tier downlink UDNs. | SIC detection scheme in clustering scenario based on an interference graph. | Maximize the average capacity and spectrum efficiency. | The average capacity of the proposed system was decreased significantly due to the number of FBS users in overlapping areas. | [138] |

**Table 4.** *Cont.*

| Issue | Methodologies | Advantages | Limitations/Future Work | Ref. |
|---|---|---|---|---|
| Increase the network capacity while considering frequency reuse usage and inter-cluster interference in UDN environment. | K-mean clustering algorithm. | Maximize the network capacity and mitigate inter-cluster interference. | When the number of clusters increased, the interference also increased, and this led to a decrease in the channel capacity of the proposed system. | [139] |
| Minimize ICI by resource allocation optimization between users in UDNs. | Cross-tier cooperation load-adapting interference management distributed strategy. | Enhance SE, SBSs throughput, EE, and ICI allocation. | Increasing the number of users who share the same bandwidth decreased the density of SBSs, which decreased the user SINR. | [140] |
| Decrease co-tier interference and maximize the data rate of the system in the UD-SCNs scenario. | -Hierarchical clustering algorithm (HCA) with distinctive forms and a hierarchical clustering approach. | Suitable for hyper-dense deploying networks of SBSs. | The effect of CCI in this technique was found to be severe due to the sub-channel being allocated to more than one SUE, resulting in data rate degradation in the proposed system. | [141] |
| Mitigate the ICI and optimize energy efficiency in the uplink mm-wave UDN multicarrier system. | Noncooperative game theory-based interference mitigation strategy in Low-complexity stair water-filling (SWF) scenario. | Improve EE performance while maintaining an acceptable SE performance with low computational complexity. | Each SUE selected its own PA strategy depending on the assumption of optimizing its EE without considering the influence of other SUEs, resulting in maximizing power consumption. | [142] |
| Mitigate ICI, accommodate additional UEs, and decrease the outage ratio of the system. | Completely distributed self-learning interference minimization (SLIM) scenario for independent networks. | Mitigate the interference and reduce power consumption while guaranteeing UEs' QoS for autonomous UDNs. | By increasing the number of users, the ICI increased as well, and the system becomes overloaded, leading to maximizing the outage ratio. | [143] |
| Optimize a UDN's downlink throughput. | Novel metric technique with an appropriate degree of Special Spectrum Reuse (SSR). | Achieving an optimal trade-off between reusing the spectrum and interference to provide high throughput. | When increasing the number of SCs, the outage threshold increased. This resulted in a decrease in the throughput. | [144] |
| Address traffic uncertainty while achieving the trade-off between load balancing and the probability of constraint in UDNs. | Chance constraint programming (CCP) with distributed sub-optimal user association and BS activation strategy based on the Markov approximation framework. | Capable of suppressing severe interference and using the density of BSs to achieve superior load balancing performance. | A significant time delay could be observed for the proposed system if there is a high traffic level. | [145] |
| Minimize the effect of interference while maximizing the SINR in SC-UDN. | A user-centric adaptive small-cell (SC) clustering strategy relying on an enhanced K-means algorithm. | Capable of dynamically adjusting the number and size of SC clusters in response to the user's SINR and effectively reducing the complexity. | An efficient radio resource allocation scheme based on a clustering strategy for UDNs could be designed. | [146] |

## 4.4. Unmanned Arial Vehicle (UAV)

A UAV, sometimes known as a drone, is a type of flying aircraft that can be controlled from the ground without the use of a human pilot. The primary application of UAVs is as temporary flying BS in B5G communication. UAVs often fly via low-reliability point-to-point connectivity, which leads to lost signal at any moment during the flight. High

reliability and minimal latency are two advantages of using a B5G network for UAV operations. This implies that a UAV can quickly receive and respond to orders sent by the ground control system or pilot. B5G speeds up the process of transmitting, receiving, and responding to orders, thus lowering the error margin that may occur during the flight. This low latency is very important when UAVs are flying in areas where a global positioning system (GPS) is not available or when they are flying beyond line of sight (BLOS). UAVs cannot use GPS in this situation, so they have to use visual-inertial odometry (VIO) to navigate in places where the view of the pilot is occluded [147]. To provide the pilot with a precise view of where the UAV is, B5G will allow the UAV's camera feed to be updated in real-time on the ground control system (GCS) of the pilot [148]. Air inspection, delivery, film and entertainment, critical missions, surveillance, intelligence, and mapping are just a few of the applications that will benefit from UAVs running on a B5G network [149]. UAVs enabled-5G can safely carry medical supplies such as COVID-19 testing to impacted populations by restricting human-to-human interaction and therefore avoiding infection spread [150]. A UAV used for rescue and search can send data and images in real-time and with low latency, which increases the speed and efficiency of the search and rescue process. In general, the accuracy and low latency of B5G will enable these new use cases and boost the adoption of UAVs. This exponential growth of UAV-enabled applications in the high-speed wireless communication field B5G has resulted in a paradigm shift in the wireless communication field [151]. The main benefit of UAV-assisted wireless communication is that it is the most appropriate technique for providing wireless connection and coverage to end-users who lack infrastructure coverage. The main benefit of UAV-assisted wireless communication is that it is the most appropriate technique for providing wireless connection and coverage to end-users who lack infrastructure coverage because of mountainous terrain, densely populated areas, severe shadowing, and degradation of communication infrastructure due to natural disasters [152]. The general scenario of UAV communication is depicted in Figure 13.

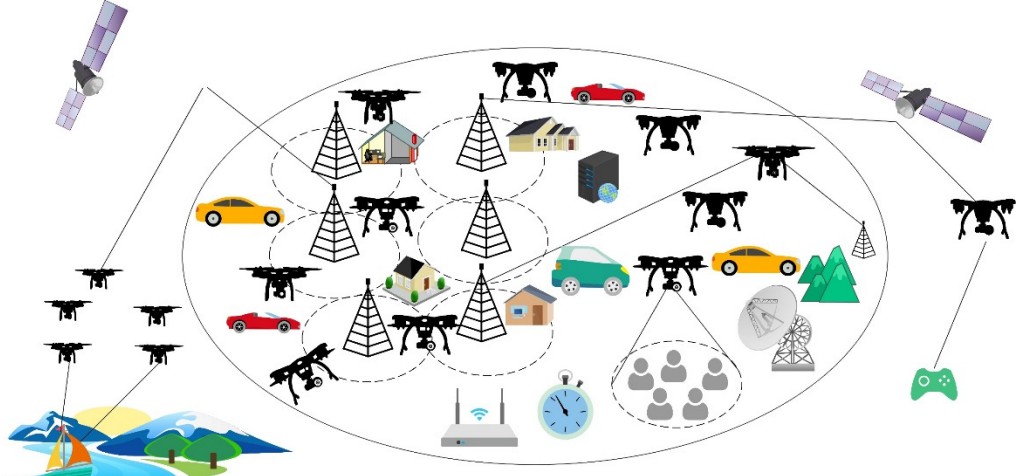

**Figure 13.** The General Scenario of UAV communication.

### 4.4.1. Unique Features of UAV

UAV communications demonstrate the following significant characteristics as a prospective option to replace or supplement terrestrial cellular networks [153,154]:

1.  LoS connections: UAVs flying in space without human pilots have a greater chance of connecting to ground users via LoS connections, which enables very reliable communications over long distances. Furthermore, UAVs can change their hovering places to preserve communication quality.
2.  The capability of dynamic deployment: In comparison to the ground station's infrastructure, UAVs can be distributed dynamically based on real-time requirements,

making them more resistant to changes in the environment. Furthermore, UAVs as aerial BSs do not need the expense of site rental, eliminating the necessity for cables and towers.

3. Swarm networks based on UAVs: A swarm of UAVs can establish scalable multi-UAV networks and provide ubiquitous connections to ground users. A multi-UAV network is a good choice for quickly restoring and expanding connectivity because it has a high degree of flexibility and speed of service.

### 4.4.2. Types of UAV

There are several varieties of UAVs. To maximize the efficiency of UAV use, it is necessary to utilize an application-particular type. UAVs are categorized based on their altitude and the type of wings [155]. Figure 14 depicts the classification of UAVs.

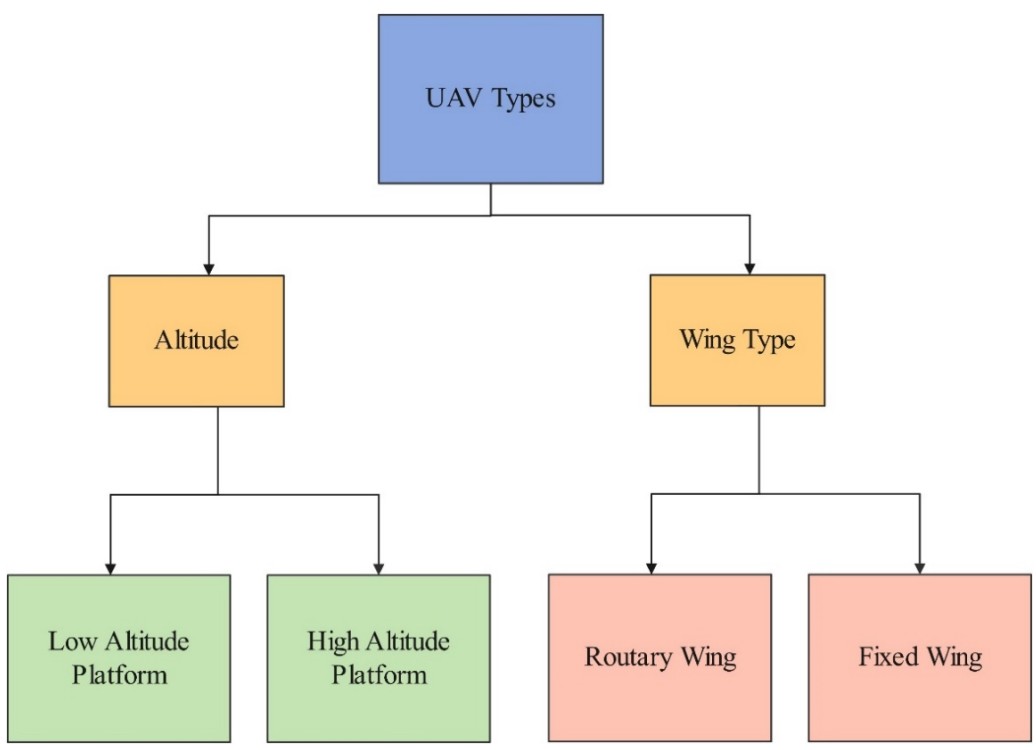

**Figure 14.** Classification of UAV.

UAVs are classified as low-altitude or high-altitude UAVs as follows:

1. Low-altitude platforms (LAPs) are easier to install and deploy than high-altitude platforms, but their coverage area is smaller, and their endurance time is shorter than high-altitude platforms.
2. High altitude platforms (HAPs) can support the task for many months, but they are more expensive to deploy than low altitude platforms (LAPs).

Depending on the wing type, UAVs can be classified as fixed-wing or rotary-wing as follows:

1. A fixed-wing creates lift utilizing forward-moving wings. It requires a runway for takeoff and landing, and it must be able to maintain a certain forward speed. Its features are simple construction, high speed, and large cargo.
2. A rotary wing uses blades that revolve around a rotor shaft to generate lift. It is capable of hovering and moving in every direction. Its mechanism depends on vertical takeoff and landing. Its features are a lower payload, a shorter range, and a slower speed [156].

### 4.4.3. Interference in UAVs

Interference with a UAV in flight may prove damaging to the UAV's mission success. The most serious types of interference are those that affect global navigation satellite system (GNSS) transmissions. This may force the UAV to compromise on the quality and accuracy of the data it stores. Once the data is analyzed, this may lead to re-fly the task again. Interference could lead to a complete loss of signal and UAV because it will lose tracking and positioning [157]. There are two main types of interference in UAVs. The first type is internal interference which represents the interference from other electronic devices on the UAV. Due to the compact size of electronic devices, certain GNSS antennas are located next to other electronic and electrical equipment. The second type is external interference which refers to the other interference sources that can come from the UAV itself, whether deliberate or not.

Certain operations, such as inspecting bridges and other relevant structures, employ UAVs that are close to roads. In this situation, the probability of interference from in-car devices such as jammers increases. This type of device is illegal, inexpensive, and easily available [158].

A UAV designer should consider utilizing receivers and antennas that are very precise to achieve very high accuracy and to get rid of interfering signals from third parties, thereby providing data with high reliability [159]. Interference in UAVs is depicted in Figure 15.

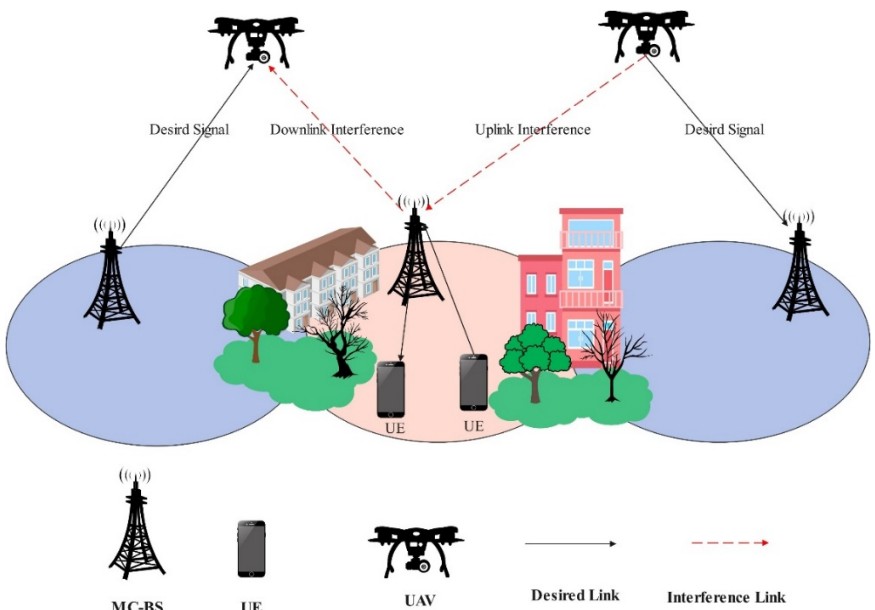

**Figure 15.** Interference in a UAV scenario.

### 4.4.4. Related Work in UAVs

In this subsection, we present the most recent studies conducted on drone interference schemes ([160–162]), inter-cell interference schemes ([163–166]), co-channel interference schemes ([167–170]), and mutual interference schemes ([171–173]) in UAVs, based on the interference management schemes used for mitigating these types of interference, as follows:

a.    Drone Interference Schemes

In [160], a reverse frequency allocation (RFA) scheme with decoupled association (DeCA) was proposed to minimize the effect of ICI, drone interference (DI) and enhance the uplink SIR of MBS coverage edge users. The DI is a result of excessive drone utilization (EDU) for 5G-enabled apps, whereas the ICI is a because of the deployment of multi-tier. Two-tier HetNet was considered in this study. Simulation results stated that the proposed scheme produces an increase in the SE due to improved uplink coverage as opposed to

the coupled association (CA) with RFA. Nonetheless, an increase in the density of drones caused significant DI and consequently reduced the UL coverage of the proposed model.

In [161], Drones were used as air routers to construct a LAN in complicated pipeline networks. An optimum 3D drone scheme based on two-phase evolution was presented for use in the deployment of drones, which allows pipeline inspectors to receive instructions and respond to accidents in real-time, thereby reducing the effect of accidents when they carry special communication equipment to inspect pipelines. The proposed scheme analyzed the quality of coverage issue and signal interference issue in two phases and minimized drone signal interference while preserving the feasible quality of coverage. The simulation results revealed that the proposed scheme can find out the optimum and maximum drone deployment in a few steps and was more feasible than the clustering and greedy schemes. However, when the number of drones and the distance between them increase, the effect of interference also increased. As a result, the spectrum efficiency of the proposed scheme decreased.

In [162], the potential benefits of joint detection (JD) in a hybrid-duplex (HD) UAV communication system were investigated as a step toward overcoming the scarcity of spectrum in UAV communications. A new method for obtaining closed-form explanations for the outage probability and limited SNR diversification gain of a joint detector operating across Rician fading channels was presented. The analysis of the multiplexing gain region (MGR) demonstrated that the HBD-UCS based on JD provides higher SNR diversification gain with better QoS requirements compared with the HBD-UCS based on interference ignorant and HBD-UCS based on the SIC detector. However, increasing the inter-UAV interference resulted in maximizing the probability of an outage. Thus, the optimal coverage probability of the proposed system decreased.

b.    Inter-cell Interference Schemes

In [163], the interference management scheme based on UAVs was proposed for optimizing the performance of in-band UAV-aided integrated access and backhaul (IAB) networks. Two modes of spatial configuration for UAVs were presented, namely distributed UAVs and drone antenna array (DAA); according to the spatial distribution of the ground user. The simulation results indicated that the attainable performance benefits are directly proportional to the number of drone elements in DAA. Moreover, the complexity of the proposed scheme was unaffected by the number of UAVs when designed as DAA. Nevertheless, when the number of UAVs increased, the mutual interference levels between access and backhaul links also increase, and this led to a decrease in the performance of the proposed scheme.

In [164], a generalized side-lobe mitigation strategy applicable to collaboration beamforming (CBF) in 3D-UAVs wireless sensor networks utilizing the gravitational search scheme (GSS) was designed to minimize interference and improve coverage capacity. The simulation results stated that the proposed strategy outperforms the peak side-lobe mitigation strategy in terms of the total side-lobe level and the performance capacity. The proposed side-lobe mitigation strategy was an excellent candidate for implementation in CBF in feasible wireless sensor networks based on 3D-UAVs. However, the mobility of sensor nodes that affect the system power consumption was neglected in this strategy.

In [165], the authors investigated a distribution algorithm for interference management in UAV HetNets two-tier downlink scenarios that influence the mobility of UAV, optimized ICIC and cell range expansion (CRE) methods. The simulation results demonstrated that a simple heuristic-based ICIC strategy beats the deep Q-learning-based ICIC strategy. Taking advantage of various optimization factors for interference coordination, the ICIC strategy based on heuristic can realize 5pSE values that are relatively close to those obtained with comprehensive brute force search strategies, with significantly less complexity. Yet, the impact of Rician or Rayleigh fading that causes a decrease in the SE of the proposed scenario was neglected.

In [166], the authors investigated interference management based on an artificial intelligence (AI) solution and used a single AI agent to model all MBSs and UABSs in the

UAV HetNet two-tier downlink scenario. A greedy algorithm and an algorithm based on double deep Q-learning (DDQN) were proposed to compute the optimal FeICIC and eICIC criteria independently for all MBSs and UABSs, and the positions of UABSs, to optimize the mean and median SE. In comparison to traditional optimization methods, for the suggested algorithms, the greedy algorithm was able to obtain better performance in terms of mean and median SE, while the AI approach achieved 95.83 % and 93.46 % of the optimum mean and median SE respectively. However, the effect of Rician or Rayleigh fading that minimizes the SE of the proposed model was not taken into consideration.

c.  Co-channel Interference Schemes

In [167], the co-channel interference in the industrial, scientific, and medical (ISM) frequency band 2.4 GHz between UAV and connected WLAN vehicles system was investigated according to the disturbing system's received SNR. The simulation results stated that if the height separation between the UAV and the WLAN-connected vehicles system is more than 6.2 km, the UAV elevation angle can guarantee that there is no co-channel interference with the WLAN-connected vehicles system in all statuses. Moreover, it was found that increasing the UAV elevation angle can mitigate the UAV interference with the WLAN-connected vehicles system. Nonetheless, this study neglected the variable altitude and elevation of the UAV that affect the received SNR of the proposed system.

In [168], the authors investigated the joint unmanned aerial vehicles-ground user (UAV-GU) association, sub-channel allocation, and UAV track control issue for wireless networks based on UAVs with spectrum re-utilization and interference management to enhance the fairness of resource participation among ground users concerning the requests of their data transition and spectrum re-utilization. Furthermore, it was demonstrated that the deployed number of UAVs, the number of subchannels, and the maximum velocity of the UAV all have significant effects on the realized maximum–minimum average rate. Nevertheless, when the number of ground users and UAVs increased, the co-channel interference also increases, and this resulted in a decrease in the average data rate of the proposed method.

In [169], the authors investigated the downlink interference problem of the UAV and internet of vehicles (IoVs) cooperative network that exists in the same region. In which the compatibility of frequency between UAVs and IoVs operating in the Ka-band is studied to quantify the separation distance necessary to avoid co-channel interference between UAVs and IoVs. The simulation results stated that the noise and interference ratio between UAV and IoV is calculated under various steering angles between interference and interfered antenna and various altitude angles from UAV to geosynchronous orbit (GSO) satellite. Yet, the interference probability and duration that have a massive effect on the system gain were not considered in the calculation process of the proposed algorithm.

In [170], the authors investigated the interference management in uplink wireless UAV-enabled information collection from a dispersed set of sensors in the scenario of IoT by using the mobility of several UAVs operating in the same band of frequency for the supporting network. According to the simulation results, the proposed algorithm with RA, and track optimization took at least 25% less time than previous dynamic orthogonal benchmark algorithms when deployed with four UAVs. Finally, a perceptible metric and associated concept for assessing the appropriateness of the suggested algorithm were presented, which can aid in the creation of a strategy for calculating the maximum number of UAVs that can be used in practice. However, this work did not take into consideration the altitude freedom of UAVs that can help to minimize interference and optimize the performance of the proposed system.

d.  Mutual Interference Schemes

In [171], the idea of associating SCs with network flying platforms (NFPs) in HetNet, eliminating total interference and maintaining a minimum data rate requirement was formulated. The simulation results stated that the proposed algorithms return sub-optimal solutions with less complexity and minimum overall interference. However, in

this study, the mobility and power consumption of NFPs, which affect the system's power consumption, were not considered.

In [172], the authors investigated a transmission method based on TDMA in both up and downlinks scenarios to maximize the data rate between a BS and UE by using multiple relaying UAVs. A joint optimum strategy for 3D track design and power allocation was proposed to maximize the network's data rate while meeting the interference restriction. The simulation results demonstrated the effectiveness of the proposed strategy in optimizing the maximum flow and mitigating interference. Yet, in this strategy, the effect of the non-LoS path that has a massive effect on the system's data rate was neglected.

In [173], a power optimization based on CoMP and clustering strategy was designed in a UAV-assisted network. The strategies of power allocation were then updated frequently until convergence is achieved, and the ultimate optimum PA result is acquired. The results demonstrated that the cluster size regularity, cluster number, and the limitation of minimum distance have large effects on interference mitigation. Nonetheless, when the number of ground users increased, the inter-cluster interference also increased, and this caused system data rate degradation. Furthermore, the mobility of ground users and UAVs that affect the power allocation of the proposed system was not taken into consideration.

The summary of previous studies of UAVs is shown in Table 5

**Table 5.** The summary of previous studies of UAVs in the literature.

| Issue | Methodologies | Advantages | Limitations/Future Work | Ref. |
|---|---|---|---|---|
| Minimize the effect of ICI, and DI and enhance the uplink SIR of MBS coverage edge users. | Reverse frequency allocation (RFA) scheme with decoupled association (DeCA). | Increase the SE. | An increase in the density of drones caused significant DI and consequently reduced the UL coverage of the proposed model. | [160] |
| Maximize signal coverage and mitigate interference in complex pipeline networks. | Optimum 3D drone scheme based on two-phase evolution. | The ability to find the optimum and maximum drone deployment in a few steps. | when the number of drones and the distance between them increased, the effect of interference also increased. As a result, the spectrum efficiency of the proposed scheme decreased. | [161] |
| Overcome the scarcity of spectrum in UAVs communications and increase SNR to mitigate inter-UAVs interference as well as eliminate outage probability. | Joint detection (JD) in a hybrid-duplex (HD) UAV communication system. | Maximize SNR diversification gain with better QoS requirements. | Increasing the inter-UAV interference resulted in increasing the probability of an outage. Thus, the optimal coverage probability of the proposed system decreased. | [162] |
| Optimize the performance of in-band UAV-aided integrated access and backhaul (IAB) networks. | Two modes of spatial configuration for UAVs were presented, namely distributed UAVs and drone antenna array (DAA). | Realize an average of 3.1X and 6.7X gains in DL SINR received signal and total sum rate as compared with the baseline scheme. | When the number of UAVs increased, the mutual interference levels between access and backhaul links also increased, and this caused a decrease in the performance of the proposed scheme. | [163] |
| Minimize interference and improve coverage capacity in 3D-UAVs wireless sensor networks. | A generalized side-lobe mitigation strategy applicable to collaboration beamforming (CBF) in 3D-UAVs wireless sensor networks | Minimize the side-lobe level and maximize the performance capacity of the networks. | The mobility of sensor nodes could be considered. | [164] |

**Table 5.** *Cont.*

| Issue | Methodologies | Advantages | Limitations/Future Work | Ref. |
|---|---|---|---|---|
| Calculate the optimal FeICIC and eICIC criteria independently for all MBSs and UABSs in LTEA HetNets based on UAV. | A dedicated sequential algorithm and an algorithm based on deep Q-learning. | Increase the fifth percentile spectral efficiency (5pSE) with significantly less complexity. | The impact of Rician or Rayleigh fading could be considered in this scenario. | [165] |
| Compute the optimal FeICIC and eICIC criteria independently for all MBSs and UABSs, and the positions of UABSs in HetNets based on UAV. | A deep Q-learning (DQN) based-greedy algorithm. | Achieve the optimum mean and median SE. | The AI approach failed to locate the optimal solution and was always surpassed by the greedy algorithm. | [166] |
| Minimize the Co-Channel Interference between UAV and WLAN-connected vehicles system. | Interference scheme generated by UAV and satellite communication on the co-channel WLAN-connected vehicles system. | Minimize the co-channel interference. | The variable altitude and elevation of the UAV that affect the received SNR of the proposed system were not considered. | [167] |
| Improve the maximum–minimum average rate under restrictions of data demand for ground users | The joint unmanned aerial vehicles-ground user (UAV-GU) association, sub-channel allocation, and UAV track control issue. | Maximize the data rate gain. | When the number of ground users and UAVs increased, the co-channel interference also increased, and this resulted in a decrease in the average data rate of the proposed method. | [168] |
| Avoid downlink co-channel interference between UAVs and IoVs Network operating in Ka-Band. | The compatibility of frequency between UAVs and IoVs operating in the Ka-band. | Minimizes the interference and noise ratio. | The interference probability and duration could be considered for the proposed algorithm. | [169] |
| Grant the wireless network with extra system gain, resilience, and sturdiness in UAVs-track design. | A general joint RA and formulation of track optimization. | Minimize the time completion for the information collected in the wireless network. | The impact of altitude could be investigated from the aspects of communication demands and EE. | [170] |
| Solve the association issue for mitigating the overall interference of the system while attaining an overall sum rate target (MITTSR). | Integer linear programming (ILP) approach by using the Gurobi optimization tool with a centralized resource allocation algorithm. | Obtain the sub-optimal solutions with less complexity and minimum overall interference. | The mobility and power consumption of NFPs, which affect the system's power consumption, were not considered. | [171] |
| Maximize the network's data rate and avoid the interference produced by reckless and smart interferences. | A joint optimum strategy for 3D track design and power allocation. | Optimize the maximum flow. | The impact of the non-LoS path was not considered. | [172] |
| Improve the UAVs' power allocation and increase the data rate by minimizing intra-cluster interference. | It is focusing on a power control strategy based on game theory and an affection propagation-assisted UAV clustering strategy using APC. | Improve the system sum rate significantly and minimize interference and avoids cluster formation. | When the number of ground users increased, the inter-cluster interference also increased, and this caused system data rate degradation. | [173] |

## 5. Comparison of B5G Networks Architecture Considering Different Types of Interference along with Critical Parameters

The architecture of B5G networks such as HetNet; D2D; UDN; and UAV have some common interference issues such as co-tier interference, cross-tier interference, inter-cell interference, intra-cell interference, inter-cluster interference, intra-cluster interference, etc. It is essential to conduct a comparison that focuses specifically on the interference in this architecture. Table 6 shows a detailed comparison of different strategies considering various types of interference. It is obvious that co-tier interference, cross-tier interference, and inter-cell interference are the most common types discussed in the interference mitigation issues, and the inter-user interference, drone interference, and mutual interference are the opposite and are rarely mentioned in the literature.

The interference level in these architecture networks is highly affected by some critical parameters. Figure 16 shows the relationship between interference level and critical parameters with respect to different architecture networks. In HetNets, the interference level increases when the number of layers increases. In D2D, an increase in the distance leads to an increase in the interference level. Moreover, the increment in the number of users in UDNs results in an increase in the interference level. In the case of UAVs, the interference level increases with the increase in the number of drones.

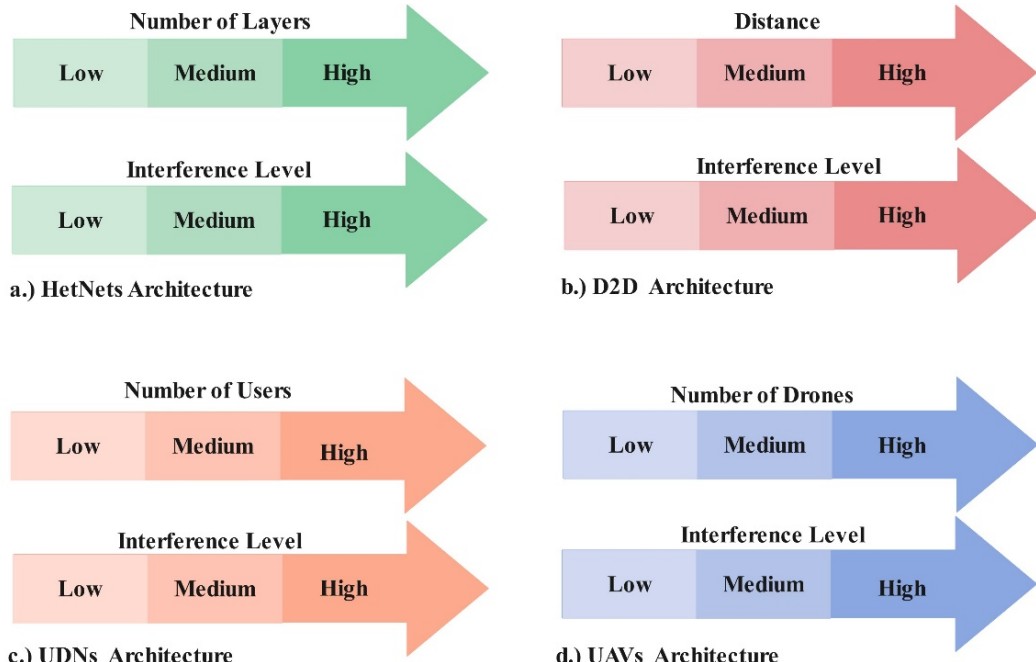

**Figure 16.** The relationship between interference level and critical parameters with respect to different architecture networks.

**Table 6.** The comparison of various strategies with respect to different types of interference.

| Architecture | References | Co-Tier Interference | Cross-Tier Interference | Inter-Cell Interference | Intra-Cell Interference | Inter-Relay Interference | Intra-Relay Interference | Inter-Cluster Interference | Intra-Cluster Interference | Co-Channel Interference | Inter-Beam Interference | Intra-Beam Interference | Inter-User Interference | Intra-User Interference | Inter-Tire Interference | Intra-Tier Interference | Drones Interference | Mutual Interference |
|---|---|---|---|---|---|---|---|---|---|---|---|---|---|---|---|---|---|---|
| HetNets | [68] | ✔ | ✗ | ✔ | ✗ | ✗ | ✗ | ✗ | ✗ | ✗ | ✗ | ✗ | ✗ | ✗ | ✗ | ✗ | ✗ | ✗ |
| | [69] | ✔ | ✗ | ✗ | ✔ | ✗ | ✗ | ✗ | ✗ | ✗ | ✗ | ✗ | ✗ | ✗ | ✗ | ✗ | ✗ | ✗ |
| | [70] | ✗ | ✗ | ✔ | ✔ | ✗ | ✗ | ✗ | ✗ | ✗ | ✗ | ✗ | ✗ | ✗ | ✗ | ✗ | ✗ | ✗ |
| | [71] | ✔ | ✗ | ✗ | ✗ | ✗ | ✗ | ✗ | ✗ | ✗ | ✗ | ✗ | ✗ | ✗ | ✗ | ✗ | ✗ | ✗ |
| | [72] | ✔ | ✗ | ✗ | ✗ | ✔ | ✗ | ✗ | ✗ | ✗ | ✗ | ✗ | ✗ | ✗ | ✗ | ✗ | ✗ | ✗ |
| | [73] | ✔ | ✗ | ✗ | ✗ | ✗ | ✗ | ✗ | ✗ | ✗ | ✗ | ✗ | ✗ | ✗ | ✗ | ✗ | ✗ | ✗ |
| | [74] | ✗ | ✗ | ✗ | ✗ | ✗ | ✗ | ✔ | ✔ | ✗ | ✗ | ✗ | ✗ | ✗ | ✗ | ✗ | ✗ | ✗ |
| | [75] | ✗ | ✔ | ✗ | ✗ | ✗ | ✗ | ✗ | ✗ | ✗ | ✗ | ✗ | ✗ | ✗ | ✗ | ✗ | ✗ | ✗ |
| | [76] | ✔ | ✔ | ✗ | ✗ | ✗ | ✗ | ✗ | ✗ | ✗ | ✗ | ✗ | ✗ | ✗ | ✗ | ✗ | ✗ | ✗ |
| | [77] | ✗ | ✔ | ✗ | ✗ | ✗ | ✗ | ✗ | ✗ | ✗ | ✗ | ✗ | ✗ | ✗ | ✗ | ✗ | ✗ | ✗ |
| | [78] | ✗ | ✔ | ✔ | ✗ | ✗ | ✗ | ✗ | ✗ | ✔ | ✗ | ✗ | ✗ | ✗ | ✗ | ✗ | ✗ | ✗ |
| | [79] | ✔ | ✔ | ✗ | ✗ | ✗ | ✗ | ✔ | ✗ | ✗ | ✗ | ✗ | ✗ | ✗ | ✗ | ✗ | ✗ | ✗ |
| | [80] | ✔ | ✔ | ✗ | ✗ | ✗ | ✗ | ✗ | ✗ | ✗ | ✗ | ✗ | ✗ | ✗ | ✗ | ✗ | ✗ | ✗ |
| D2D | [98] | ✗ | ✗ | ✗ | ✗ | ✗ | ✗ | ✗ | ✗ | ✗ | ✔ | ✗ | ✗ | ✗ | ✗ | ✗ | ✗ | ✗ |
| | [99] | ✔ | ✔ | ✗ | ✗ | ✗ | ✗ | ✗ | ✗ | ✗ | ✔ | ✔ | ✗ | ✗ | ✗ | ✗ | ✗ | ✗ |
| | [100] | ✔ | ✔ | ✗ | ✗ | ✗ | ✗ | ✗ | ✗ | ✗ | ✗ | ✗ | ✗ | ✗ | ✗ | ✗ | ✗ | ✗ |
| | [2] | ✔ | ✔ | ✗ | ✗ | ✔ | ✔ | ✗ | ✗ | ✗ | ✗ | ✗ | ✗ | ✗ | ✗ | ✗ | ✗ | ✗ |
| | [101] | ✔ | ✔ | ✗ | ✗ | ✗ | ✗ | ✗ | ✗ | ✗ | ✗ | ✗ | ✗ | ✗ | ✗ | ✗ | ✗ | ✗ |
| | [102] | ✗ | ✔ | ✗ | ✗ | ✗ | ✗ | ✗ | ✗ | ✗ | ✗ | ✗ | ✗ | ✗ | ✗ | ✗ | ✗ | ✗ |
| | [103] | ✗ | ✗ | ✗ | ✔ | ✗ | ✗ | ✗ | ✗ | ✗ | ✗ | ✗ | ✗ | ✗ | ✗ | ✗ | ✗ | ✗ |
| | [104] | ✔ | ✗ | ✗ | ✗ | ✗ | ✗ | ✗ | ✗ | ✗ | ✗ | ✗ | ✗ | ✗ | ✗ | ✗ | ✗ | ✗ |
| | [105] | ✗ | ✗ | ✗ | ✗ | ✗ | ✗ | ✔ | ✔ | ✗ | ✗ | ✗ | ✗ | ✗ | ✗ | ✗ | ✗ | ✗ |
| | [106] | ✗ | ✗ | ✗ | ✗ | ✗ | ✗ | ✔ | ✔ | ✗ | ✔ | ✔ | ✗ | ✗ | ✗ | ✗ | ✗ | ✗ |
| | [107] | ✗ | ✔ | ✗ | ✗ | ✗ | ✗ | ✗ | ✗ | ✗ | ✗ | ✗ | ✗ | ✗ | ✗ | ✗ | ✗ | ✗ |
| | [108] | ✗ | ✔ | ✗ | ✗ | ✗ | ✗ | ✗ | ✗ | ✔ | ✗ | ✗ | ✗ | ✔ | ✗ | ✗ | ✗ | ✗ |
| | [109] | ✔ | ✔ | ✗ | ✗ | ✗ | ✗ | ✗ | ✗ | ✗ | ✗ | ✗ | ✗ | ✗ | ✗ | ✗ | ✗ | ✗ |

**Table 6.** *Cont.*

| Architecture | References | Co-Tier Interference | Cross-Tier Interference | Inter-Cell Interference | Intra-Cell Interference | Inter-Relay Interference | Intra-Relay Interference | Inter-Cluster Interference | Intra-Cluster Interference | Co-Channel Interference | Inter-Beam Interference | Intra-Beam Interference | Inter-User Interference | Intra-User Interference | Inter-Tire Interference | Intra-Tier Interference | Drones Interference | Mutual Interference |
|---|---|---|---|---|---|---|---|---|---|---|---|---|---|---|---|---|---|---|
| UDNs | [133] | ✔ | ✔ | ✗ | ✗ | ✗ | ✗ | ✗ | ✗ | ✗ | ✗ | ✗ | ✗ | ✗ | ✗ | ✗ | ✗ | ✗ |
| | [134] | ✗ | ✗ | ✔ | ✗ | ✗ | ✗ | ✗ | ✗ | ✗ | ✗ | ✗ | ✗ | ✗ | ✗ | ✗ | ✗ | ✗ |
| | [135] | ✗ | ✗ | ✗ | ✗ | ✗ | ✗ | ✗ | ✗ | ✗ | ✗ | ✗ | ✔ | ✗ | ✔ | ✗ | ✗ | ✗ |
| | [136] | ✗ | ✗ | ✔ | ✗ | ✗ | ✗ | ✗ | ✗ | ✗ | ✗ | ✗ | ✗ | ✗ | ✗ | ✗ | ✗ | ✗ |
| | [137] | ✗ | ✗ | ✗ | ✗ | ✗ | ✗ | ✗ | ✗ | ✔ | ✗ | ✗ | ✔ | ✗ | ✗ | ✗ | ✗ | ✗ |
| | [138] | ✔ | ✔ | ✗ | ✗ | ✗ | ✗ | ✗ | ✗ | ✗ | ✗ | ✗ | ✗ | ✗ | ✗ | ✗ | ✗ | ✗ |
| | [139] | ✗ | ✗ | ✔ | ✗ | ✗ | ✗ | ✗ | ✗ | ✗ | ✗ | ✗ | ✗ | ✗ | ✗ | ✗ | ✗ | ✗ |
| | [140] | ✗ | ✔ | ✔ | ✗ | ✗ | ✗ | ✗ | ✗ | ✗ | ✗ | ✗ | ✗ | ✗ | ✗ | ✗ | ✗ | ✗ |
| | [141] | ✔ | ✗ | ✗ | ✗ | ✗ | ✗ | ✗ | ✗ | ✗ | ✗ | ✗ | ✗ | ✗ | ✗ | ✗ | ✗ | ✗ |
| | [142] | ✗ | ✗ | ✔ | ✗ | ✗ | ✗ | ✗ | ✗ | ✗ | ✗ | ✗ | ✗ | ✗ | ✗ | ✗ | ✗ | ✗ |
| | [143] | ✗ | ✗ | ✔ | ✗ | ✗ | ✗ | ✗ | ✗ | ✗ | ✗ | ✗ | ✗ | ✗ | ✗ | ✗ | ✗ | ✗ |
| | [144] | ✗ | ✗ | ✔ | ✗ | ✗ | ✗ | ✗ | ✗ | ✗ | ✗ | ✗ | ✗ | ✗ | ✗ | ✗ | ✗ | ✗ |
| | [145] | ✗ | ✗ | ✔ | ✗ | ✗ | ✗ | ✗ | ✗ | ✗ | ✗ | ✗ | ✗ | ✗ | ✗ | ✗ | ✗ | ✗ |
| | [146] | ✗ | ✗ | ✗ | ✗ | ✗ | ✗ | ✗ | ✗ | ✗ | ✗ | ✗ | ✗ | ✗ | ✔ | ✔ | ✗ | ✗ |
| UAVs | [160] | ✗ | ✗ | ✔ | ✗ | ✗ | ✗ | ✗ | ✗ | ✗ | ✗ | ✗ | ✗ | ✗ | ✗ | ✗ | ✔ | ✗ |
| | [161] | ✗ | ✗ | ✗ | ✗ | ✗ | ✗ | ✗ | ✗ | ✗ | ✗ | ✗ | ✗ | ✗ | ✗ | ✗ | ✔ | ✗ |
| | [162] | ✗ | ✗ | ✗ | ✗ | ✗ | ✗ | ✗ | ✗ | ✗ | ✗ | ✗ | ✗ | ✗ | ✗ | ✗ | ✔ | ✗ |
| | [163] | ✗ | ✔ | ✔ | ✗ | ✗ | ✗ | ✗ | ✗ | ✗ | ✗ | ✗ | ✗ | ✗ | ✗ | ✗ | ✔ | ✗ |
| | [164] | ✗ | ✗ | ✔ | ✗ | ✗ | ✗ | ✗ | ✗ | ✗ | ✗ | ✗ | ✗ | ✗ | ✗ | ✗ | ✗ | ✗ |
| | [165] | ✗ | ✗ | ✔ | ✗ | ✗ | ✗ | ✗ | ✗ | ✗ | ✗ | ✗ | ✗ | ✗ | ✗ | ✗ | ✗ | ✗ |
| | [166] | ✗ | ✗ | ✔ | ✗ | ✗ | ✗ | ✗ | ✗ | ✗ | ✗ | ✗ | ✗ | ✗ | ✗ | ✗ | ✗ | ✗ |
| | [167] | ✗ | ✗ | ✗ | ✗ | ✗ | ✗ | ✗ | ✗ | ✔ | ✗ | ✗ | ✗ | ✗ | ✗ | ✗ | ✗ | ✗ |
| | [168] | ✗ | ✗ | ✗ | ✗ | ✗ | ✗ | ✗ | ✗ | ✔ | ✗ | ✗ | ✗ | ✗ | ✗ | ✗ | ✗ | ✗ |
| | [169] | ✗ | ✗ | ✗ | ✗ | ✗ | ✗ | ✗ | ✗ | ✔ | ✗ | ✗ | ✗ | ✗ | ✗ | ✗ | ✗ | ✗ |
| | [170] | ✗ | ✗ | ✗ | ✗ | ✗ | ✗ | ✗ | ✗ | ✔ | ✗ | ✗ | ✗ | ✗ | ✗ | ✗ | ✔ | ✗ |
| | [171] | ✗ | ✗ | ✗ | ✗ | ✗ | ✗ | ✗ | ✗ | ✗ | ✗ | ✗ | ✗ | ✗ | ✗ | ✗ | ✗ | ✔ |
| | [172] | ✗ | ✗ | ✗ | ✗ | ✗ | ✗ | ✗ | ✗ | ✗ | ✗ | ✗ | ✗ | ✗ | ✗ | ✗ | ✗ | ✔ |
| | [173] | ✗ | ✗ | ✗ | ✗ | ✗ | ✗ | ✗ | ✗ | ✗ | ✗ | ✗ | ✗ | ✗ | ✗ | ✗ | ✔ | ✔ |

## 6. Open Issue and Future Direction

### 6.1. HetNets

Future HetNets will introduce other sources of interference in addition to SI, complicating SI management and FD transmission implementation. This problem is increased in a multi-tier HetNet or multi-cell hierarchical system as the number of SCs increases. Furthermore, there are several sources of ICI, UE-UE interference, and BS-BS interference when the end-user has an FD transmission enabled with a reuse factor of one. In this situation, the management of radio resources becomes highly complex and difficult. However, a probability of increasing SE is possible. This can be realized by utilizing an effective power allocation and radio RA strategy. The practical balancing between the performance of the system and the SI management strategy performance is a promising new topic for future research [174].

The solutions to minimize co- and cross-tier interference are critical to ensure the HetNets total performance. To this day, interference minimization is a difficult problem in the management of resources since it must maintain the throughput, spectra, and EE of the system while still maintaining a reasonable amount of complexity. This will be an attractive research field for investigating the trade-off between suppressing interference and allocating radio resources. Future users may have diverse requirements for a wide range of applications; consequently, it is critical to develop the appropriate regulations for accessibility, radio resources impact, and management of interference in multi-tier HetNets. Another important issue that must be taken into consideration in interference minimization approaches is to mitigate the number of signals overhead that include CSI to minimize the number of information exchanges between small and macro-BSs [51].

### 6.2. D2D

One of the most important characteristics of B5G cellular networks is the use of millimeter-wave and terahertz transmission. Due to the operation of mm-wave and terahertz communication on a wider frequency spectrum (30–300 GHz) and (300 GHz–10 THz), respectively, these frequencies have the potential to deliver extraordinarily high data rates for mobile devices, potentially providing massive network capacity [7,175]. However, they have many significant propagation features that are distinct from those of the microwave band, leading to several challenges regarding interference management (IM). In B5G cellular networks enabling D2D communication, several scenarios are introduced within each cell to mitigate interference. However, novel IM strategies must be suggested that consider directional interference in B5G cellular networks to support a wide range of D2D communication [176]. On the other hand, IM is one of the main issues when D2D technology is integrated with SCs in underlay in-band D2D communication. Due to the transmission power of each BS being different, IM and RA problems for underlay spectrum sharing are more difficult in multi-tier HetNet compared with conventional single-tier systems. To increase spectrum efficiency, interference between D2D links must be taken into consideration and managed efficiently. Additionally, the mode selection strategies must be adopted in this heterogeneous environment to make dynamic decisions depending on the network's condition. As a result, it is important to investigate how to achieve efficient IM [177].

### 6.3. UDNs

Interference is one of the main challenges in UDNs because of the presence of dominating interference close to the intended receiver. Therefore, IM is one of the significant issues when UDNs are integrated with other B5G technologies. To overcome this issue, cell coordination and distributed control strategies are implemented to minimize total interference. Overhead signaling is required for cell coordination. The coordination of interference is a sophisticated challenge in UDNs because of the varying BS density, which causes dominant interference in some BSs [115]. As a result, joint IM and handover among different networks while taking into consideration SE and EE require more research. more-

over, the EE of an M-MIMO network with an FD relay channel should be investigated for the integration of FD, M-MIMO, and UDNs communication to mitigate the influence of interference [178]. Additionally, effective interference avoidance strategies are critical for the successful deployment of UDNs. Furthermore, the generalized coordinated multipoint (GCoMP) strategy is considered a viable solution for tackling the issue of interference, particularly for cell edge users [179].

### 6.4. UAVs

Advanced wireless communications will face several challenges, particularly in the UAVs, since UEs serviced by UAVs are more likely to suffer interference from terrestrial BSs. Thus, the development of interference management strategies for UAVs is still an urgent issue. Therefore, we illustrate many crucial challenges in IM for future UAVs. First, the UAVs cannot handle the high-speed communication requirements due to new traffic demands without AI function. As a result, AI is critical to the development of B5G communication. The use of AI in UAVs can enable more adaptive interference coordination and optimize SE and EE in comparison to the current fixed RA strategy. Primarily by interacting with the environment, AI solves the issues of learning variations, classifying problems, forecasting future obstacles, and identifying possible solutions. Another challenge is designing a simple interference coordination strategy based on AI that can completely match the data. The derivation of this strategy, which contains several parameters, will be challenging to interpret and certain values may even be lost during practical application. In this case, the strategy based on AI will not have any positive impact on IM for UEs served by UAVs. Additionally, the adoption of complex interference coordination models to address issues necessarily increases the computational burden on UAVs and BSs. Despite the attractive efficiency of AI in the future network, the long training time and computational complexity are still urgently challenged [180,181]. Second, the consumption of energy has become one of the critical challenges. Although RA can help to minimize the energy consumption of UAVs, several challenges remain. As a result, the developed methods to minimize interference and maximize EE for UAVs will be an important issue in future studies. Finally, although several efforts are being implemented to address the capacity needs of IM strategies in UAVs, the increasing demand for data will place additional strain on the front and backhaul links in the future.

Existing UAVs have significant difficulties in minimizing interference in polynomial time as the number of UEs supplied by UAVs and terrestrial UEs increases. Therefore, a sub-optimum solution with less complexity should be suggested for future UAVs. For the backhaul connection, with the densification of heterogeneous nodes such as BSs, remote radio heads (RRHs), and SCs in terrestrial networks, a sophisticated self-organizing function should be used to enable the terrestrial nodes to operate intelligently and autonomously. Additionally, the restricted backhaul capacity influences the performance of IM strategies, since considerable amounts of data and control signals must exchange among BSs to coordinate their operations. In the future, an efficient strategy with less complexity should be implemented to improve the performance of UAVs with non-zero latency and restricted capacity [182].

## 7. Conclusions

A wide range of devices and applications will be supported by the next generation, which will raise demand for massive data rates with almost zero latency. The system should maintain the spectrum frequency and QoS for each user. Unfortunately, severe interference in wireless networks causes the wireless links to degrade and reduces the system performance, which leads to preventing its commercial deployment. In the B5G networks, the interference issues are expected to be more critical since there is continuously growing traffic, density, and size. The developments of interference management techniques result in increasing the knowledge and ability to understand and enhance the performance of the networks. This paper presents a comprehensive review of the interference management

issues in B5G wireless networks and highlights the importance of interference issues in HetNets, D2D, UDNs and UAVs by focusing on the methodologies, strengths, limitations, and the upcoming challenges of the latest related works. Recent interference management issues in HetNets were reviewed by considering co-tier interference, cross-tier interference, and hybrid interference. Similarly, the paper addresses the issues of interference in D2D with a focus on power allocation, spectrum allocation, and hybrid strategies. Additionally, in UDNs, different approaches such as time domain, frequency domain, power domain and spatial domain were extensively discussed. Moreover, extensive discussion on the interference management in UAVs was provided by focusing on several interference schemes including drone interference, inter-cell interference, co-channel interference and mutual interference. Overall, several state-of-the-art studies along with open issues and prospective research directions related to the interference management issues in B5G were presented and discussed in this comprehensive review. This article is anticipated to serve as an effective and practical evolutionary guide for the design of next-generation wireless networks.

**Author Contributions:** Conceptualization: K.D. and K.A.N.; Methodology: O.T.H.A. and M.N.H.; Visualization: K.D. and R.H., Writing—original draft preparation: O.T.H.A. and M.N.H., Writing—review and editing: F.Q., A.N.A.W. and K.D.; Funding acquisition: F.Q. and A.N.A.W. All authors have read and agreed to the published version of the manuscript.

**Funding:** This paper is supported by the Universiti Kebangsaan Malaysia Research Grant, GGPM-2019-065.

**Institutional Review Board Statement:** Not applicable.

**Informed Consent Statement:** Not applicable.

**Data Availability Statement:** Not applicable.

**Acknowledgments:** The authors would also like to thank the respected Editor and Reviewer for their support.

**Conflicts of Interest:** The authors declare no conflict of interest.

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
