# Peer review of "Interference Challenges and Management in B5G Network Design: A Comprehensive Review"

_electronics, doi:10.3390/electronics11182842_

Round 1

Reviewer 1 Report

Interference problem will still be one of main problems in B5G networks. In this manuscript, the authors review the interference management for B5G HetNet including UDN, D2D, UAV. I suggest the authors add a review on the network architecture of B5G for reducing the interference. For example the recent work on cell-free massive MIMO is a good direction to reduce interference.

Author Response

We wish to express our gratitude to the reviewer for his insightful comments to improve this paper.

Attached is the response file.

Reviewer 2 Report

The timeliness of the presented research is very high, due to the ever-increasing rate of the wireless communication system evolution and the increasing need in interference management.

The presented literature review is up-to-date and is broad enough to cover the assumed technologies.

There are several that can help to make the submission more beneficial:

Minor issues that are mainly related to the submission formatting

1. The references are not formatted according to the MDPI Electronics submission rules.

2. It would be more informative to replace Figure 2 with a tree-like structure of the assumed interference sources in B5G networks, for example, like in Figure 1.

3. Some minor punctuation errors are present and should be fixed in the final submission.

Major issues that are mainly related to the submission structure.

4. All of the assumed cases (HetNet; D2D; UDN; UAV) have some common issues with interference. Thus the paper will greatly benefit form a cross comparison of those systems.

5. A littles light is shed on the quantitative effects of interference management in the assumed systems. The authors must append their analysis with the quantitative results; this will help to demonstrate to which state the interference can be cancelled with the described strategies.

Author Response

(The authors gave the same response as above.)

Round 2

Reviewer 2 Report

The authors have addressed my concerns and I think the submission can be accepted in the present form.